# Multivariate Statistical Modelling of Compound Events via Pair-Copula Constructions: Analysis of Floods in Ravenna (Italy)

Emanuele Bevacqua[1], Douglas Maraun[1], Ingrid Hobæk Haff[2], Martin Widmann[3], and Mathieu Vrac[4]

[1]Wegener Center for Climate and Global Change, University of Graz, Graz, Austria
[2]Department of Mathematics, University of Oslo, Oslo, Norway
[3]School of Geography, Earth and Environmental Sciences, University of Birmingham, Birmingham, United Kingdom
[4]Laboratoire des Sciences du Climat et de l'Environnement, CNRS/IPSL, Gif-sur-Yvette, France

*Correspondence to:* Emanuele Bevacqua (emanuele.bevacqua@uni-graz.at)

**Abstract.** Compound events (CEs) are multivariate extreme events in which the individual contributing variables may not be extreme themselves, but their joint - dependent - occurrence causes an extreme impact. Conventional univariate statistical analysis cannot give accurate information regarding the multivariate nature of these events. We develop a conceptual model, implemented via pair-copula constructions, which allows for the quantification of the risk associated with compound events in present day and future climate, as well as the uncertainty estimates around such risk. The model includes predictors, which could represent for instance meteorological processes, that provide insight into both the involved physical mechanisms, and the temporal variability of compound events. Moreover, this model enables multivariate statistical downscaling of compound events. Downscaling is required to extend the compound events risk assessment to the past or future climate, where climate models either do not simulate realistic values of the local variables driving the events, or do not simulate them at all. Based on the developed model, we study compound floods, i.e. joint storm surge and high river runoff, in Ravenna (Italy). To explicitly quantify the risk, we define the impact of compound floods as a function of sea and river levels. We use meteorological predictors to extend the analysis to the past, and get a more robust risk analysis. We quantify the uncertainties of the risk analysis observing that they are very large due to the shortness of the available data, though this may also be the case in other studies where they have not been estimated. Ignoring the dependence between sea and river levels would result in an underestimation of risk, in particular the expected return period of the highest compound flood observed increases from about 20 to 32 years when switching from the dependent to the independent case.

## 1 Introduction

On the $6^{th}$ of February 2015, a low pressure system that developed over the north of Spain moved across the Island of Corsica into Italy. The low pressure itself (Figure 1) and the associated southeasterly winds drove a storm surge to the Adriatic coast at Ravenna (Italy). Alongside the storm surge, large amounts of precipitation fell in the surrounding area causing high values of discharge in small rivers near the coast. These river discharges were partially obstructed from draining into the sea by the storm surge, which then contributed to major flooding along the coast.

[Figure 1 about here.]

Such a *compound flood* is a typical example of a *compound event (CE)*. CEs are multivariate extreme events in which the individual *contributing variables* may not be extreme themselves, but their joint - dependent - occurrence causes an extreme *impact*. The impact of CEs may be a climatic variable such as the gauge level (e.g. for compound floods), or other relevant variables such as fatalities or economic losses. CEs have received little attention so far, as underlined in the report of the
Intergovernmental Panel on Climate Change on extreme events (Seneviratne et al., 2012).

CEs are responsible for a very broad class of impacts on society. For example, heatwaves amplified by the lack of soil moisture, which reduces the latent cooling, may be classified as CEs (Fischer et al., 2007; Seneviratne et al., 2010). The impact of drought cannot be fully described by a single variable (e.g. De Michele et al., 2013; Shiau et al., 2007): analyses have been carried out which consider drought severity, duration (Shiau et al., 2007), maximum deficit (Saghafian and Mehdikhani, 2013),
as well as the affected area (Serinaldi et al., 2009). Another example of CE includes fluvial floods resulting from extreme rainfall occurring on a wet catchment (Pathiraja et al., 2012).

In recent literature, more attention has been given to the study of CEs through multivariate statistical methods (Seneviratne et al., 2012) which can offer more in-depth information, regarding the multivariate nature of CEs, than conventional univariate analysis. Combinations of univariate analyses for studying CEs are only sufficient when no dependence exists among the
compound variables. However this is not usually the case, and so would lead to misleading conclusions about the assessment of the risk associated with CEs.

Modelling CEs is a complex undertaking (Leonard et al., 2014), and methods to adequately study them are required. Parametric multivariate statistical models allow one to constrain the dependencies between the contributing variables of CEs, as well as their marginal distributions (e.g. Hobæk Haff et al., 2015; Serinaldi, 2015; Aghakouchak et al., 2014; Saghafian and
Mehdikhani, 2013; Serinaldi et al., 2009; Shiau et al., 2007; Shiau, 2003). The parametric structure reduces the uncertainties of the statistical properties we want to estimate from the data, compared to empirical estimates. However, such a reduction of the uncertainties depends on the choice of a proper parametric model. As observed data are often limited, the uncertainties might be substantial and should thus be quantified (Serinaldi, 2015).

Due to the complex dependence structure between the contributing variables, advanced multivariate statistical models are
necessary to model CEs. For example, modelling the multivariate probability distribution of the contributing variables with multivariate Gaussian distributions would usually not produce satisfying results. A multivariate Gaussian distribution would assume that the dependencies between all the pairs are of the same type (*homogeneity of the pair-dependencies*), and without any dependence of the extreme events, also called *tail dependence*. Furthermore, a multivariate Gaussian distribution would assume that all of the marginal distributions would be Gaussian. To solve the latter problems, the use of copulas has been
introduced in geophysics and climate science (e.g. Schölzel and Friederichs, 2008; Salvadori et al., 2007). Through copulas, it is possible to model the dependence structure of variables separately from their marginal distributions. However, multivariate parametric copulas lack flexibility when modelling systems with high dimensionality, where heterogeneous dependencies exist among the different pairs (Aas et al., 2009). Therefore, this lack of flexibility of copulas would be a limitation for many types of compound events. Pair-copula constructions (PCCs) decompose the dependence structure into bivariate copulas (some of

which are conditional) and give greater flexibility in modelling generic high-dimensional systems compared to multivariate parametric copulas (Aas et al., 2009; Acar et al., 2012; Bedford and Cooke, 2002; Hobæk Haff, 2012).

Here we develop a multivariate statistical model, based on PCCs, which allows for an adequate description of the dependencies between the contributing variables. The model provides a straightforward quantification of risk uncertainty, which is reduced with respect to the uncertainties obtained when computing the risk directly on the observed data of the impact. We extend the multivariate statistical model through including predictors for the contributing variables. Such predictors could represent for instance meteorological processes driving the contributing variables. This increase in complexity of the model due to additional variables, is accommodated for through the use of PCCs. The predictors allow us to (1) gain insight into the physical processes underlying CEs, as well as into the temporal variability of CEs, and (2) to statistically downscale CEs and their impacts. Downscaling may be used to statistically extend the risk assessment back in time to periods where observations of the predictors, but not of the contributing variables and impacts are available, or to assess potential future changes in CEs based on climate models. Based on this model we study compound flooding in Ravenna.

In the context of compound floods, the dependence between rainfall and sea level has previously been studied for other regions (e.g., Wahl et al., 2015; Zheng et al., 2013; Kew et al., 2013; Svensson and Jones, 2002; Lian et al., 2013). Among these studies, Wahl et al. (2015) observed an increase in the risk of compound flooding in major US cities driven by an increasing dependence between storm surges and extreme rainfall. The impact of compound floods can be described as the gauge level in a river near the coast, which is driven both by the river discharge upstream and the sea level. Only a few studies have explicitly quantified the impact of compound floods and the associated risks (Zheng et al., 2015, 2014; Van den Hurk et al., 2015; Van Den Brink et al., 2005). The reason might be difficulties in quantifying the impact due to a lack of data. For the Rotterdam case study, the impact has been explicitly quantified (Van Den Brink et al., 2005; Kew et al., 2013; Klerk et al., 2015). However, there is still debate as to whether the floods in this case are actually CEs, i.e., if surges and discharges can be treated independently or not when assessing the risk of flooding. As discussed in Klerk et al. (2015), a significant dependence is more likely in small catchments, such as those in mountainous areas by the coast, which have a quick response time to rainfall that may favour the coincidence of high river flows and storm surges driven by the same synoptic weather system.

Here, we explicitly define the impact of compound floods as a function of sea and river levels in order to quantify the flood risk and its related uncertainties. Moreover we quantify the risk underestimation that occurs when the dependence among sea and river levels is not considered. We identify the meteorological predictors driving the river and sea levels. By incorporating such predictors into the statistical model, we extend the analysis of compound floods into the past, where data are available for predictors, but not for the river and sea level stations.

The paper is organized as follows. The Ravenna case study is discussed in section 2. We introduce the conceptual model for compound events in section 3. Pair-copula constructions, i.e. the mathematical method we use to implement the model, are introduced in section 4. Based on the presented conceptual model for compound events, in section 5 we develop the model for compound floods in Ravenna. Results are presented in section 6, discussion and conclusions are provided in section 7. More technical details can be found in the appendices.

## 2 Compound flooding in the coastal area of Ravenna

In this study, we focus on the risk of compound floods in the coastal area of Ravenna. The choice of the case study was motivated by the extreme event that happened on the $6^{th}$ of February 2015, as presented in the introduction. On the day prior to the event, values of up to approximately 80 mm of rain were recorded in the surrounding area of Ravenna, and around 90 mm on the day of the event itself. The sea level recorded was the highest observed in the last 18 years (Arpa Emilia-Romagna, 2015). The high risk of flooding to population in the Ravenna region has been underlined by the *LIFE PRIMES* project (Life Primes, a), recently financed by the European Commission, whose target is "to reduce the damages caused to the territory and population by events such as floods and storm surges" (Life Primes, b) in Ravenna and its surrounding areas. As pointed out by Masina et al. (2015), natural and anthropogenic subsidences represent a threat for the coastal area of Ravenna, characterized by land elevation which are in many places below 2 m above mean sea level (Gambolati et al., 2002). The sea level inundation risk along the coast of Ravenna has been recently studied by Masina et al. (2015), who considered the joint effect of sea water level and significant wave height.

A schematic representation of the catchment on which we focus is shown in the black rectangle of Figure 2. The $Y$ variables, river and sea levels, represent the contributing variables, and the the water level $h$ is the impact of the compound flood. The $X$ variables are meteorological predictors of the contributing variables $Y$, which will be discussed in more detail later.

[Figure 2 about here.]

We develop a multivariate statistical model able to assess the risk of compound floods in Ravenna. Our research objectives are the following:

1. Develop a statistical model to represent the dependencies between the contributing variables of the compound floods, via pair-copula constructions.

2. Explicitly define the impact of compound floods as a function of the contributing variables. This allows us to estimate the risk and the related uncertainty.

3. Identify the meteorological predictors for the contributing variables $Y$. Incorporate the meteorological predictors in the model to gain insight into the physical mechanisms driving the compound floods and into their temporal variability.

4. Extend the analysis into the past (where data are available for the predictors, but not for the contributing variables $Y$).

### 2.1 Dataset

The data used here for the contributing variables $Y$ and the impact $h$ are water levels at a daily resolution (daily averages of hourly measurements). We use data for the extended winter season (November-March) of the period 2009-2015. Data sources are the *Italian National Institute for Environmental Protection and Research (ISPRA)* for the sea, and *Arpae Emilia-Romagna* for rivers and impact. River data were processed in order to mask periods of low quality, i.e. those suspected to be influenced by human activities such as the use of a dam. Moreover, we applied a procedure to homogenise the data of the rivers; details

are given in appendix A. We do not filter out the astronomical tide component of the sea level, considering that the range of variation of the daily average of sea level is about 1 meter, while that of the astronomical tide is about 9 cm. To check the above, we used astronomical tide obtained through FES2012, which is a software produced by *Noveltis, Legos and CLS Space Oceanography Division* and distributed by *Aviso*, with support from *Cnes* (http://www.aviso.altimetry.fr/). Meteorological
predictors were obtained from the *ECMWF ERA-Interim* reanalysis dataset (covering the period 1979-2015, with $0.75 \times 0.75$ degrees of resolution (Dee et al., 2011)). Specifically, for the river predictors we use daily data (sum of 12-hourly values) of total precipitation, evaporation, snow melt and snow fall, while for the sea level predictor we use daily data (average of 6-hourly values) of sea level pressure.

## 3   Conceptual conditional model for Compound Events

Leonard et al. (2014) define a CE as "an extreme impact that depends on multiple statistically dependent variables or events". This definition stresses the extremeness of the impact rather than that of the individual contributing variables, which may not be extreme themselves, and the importance of the dependence between these contributing variables. The physical reasons for the dependence among the contributing variables can be different. There can be a mutual reinforcement of one variable by the other and vice versa due to system feedbacks, e.g., the mutual enhancement of droughts and heat waves in transitional
regions between dry and wet climates (Seneviratne et al., 2012). Or the probability of occurrence of the contributing variables can be influenced from a large scale weather condition, as has occurred in Ravenna (Figure 1), where the low pressure system caused coinciding extremes of river runoff and sea level. It is clear then, that the dependence among the contributing variables represents a fundamental aspect of compound events, and so it must be properly modelled to represent these extreme events well.

Our statistical *conditional model* consists of three components: the contributing variables $Y_i$, including a model of their dependence structure, the impact $h$, and predictors $X_j$ of the contributing variables. The contributing variables $Y_i$ and their multivariate dependence structure drive the CE. For instance, in case of compound floods, the contributing variables are runoff and sea level. The impact $h$ of a CE can be formalized via an *impact-function* $h = h(Y_1, ..., Y_n)$. In the case of compound flooding, we define the river gauge level in Ravenna as impact, but in principle it can be any measurable variable such as
agricultural yield or economic loss. The predictors $X_j$ provide insight into the physical processes underlying CEs, including the temporal variability of CEs, and can be used to statistically downscale CEs when the variables $Y$ and the impact $h$ are available (e.g. Maraun et al., 2010).

The downscaling feature is particularly useful for compound events, which are not realistically simulated, or may not even be simulated at all by available climate models. For instance, standard global and regional climate models do not simulate
realistic runoff (Flato et al., 2013; Materia et al., 2010; Tisseuil et al., 2010), and do not simulate sea surges. Here, our model can be used to downscale these contributing variables, e.g. from simulated large-scale meteorological predictors. In particular, the model provides a simultaneous, i.e. multivariate, downscaling of the contributing variables $Y_i$, which allows for a realistic representation both of the dependencies between the $Y_i$, and of their marginal distributions. This is relevant because a separate

downscaling of the contributing variables $Y_i$ may lead to unrealistic representations of the dependencies between the $Y_i$, which in turn would cause a poor estimation of the impact $h$. The downscaling feature can be useful to extend the risk analysis into the past, where observations of the predictors, but not of the contributing variables and impacts are available.

More specifically, the conceptual conditional model consists of:

1. An impact function to quantify the impact $h$:

$$h = h(Y_1, ..., Y_n). \tag{1}$$

2. Predictors $X$ for the contributing variables $Y$.

3. A conditional joint probability density function (pdf) $f_{\boldsymbol{Y}|\boldsymbol{X}}(\boldsymbol{Y}|\boldsymbol{X})$ of the contributing variables $Y$, given the predictors $X$ (which we describe through a parametric model, via pair-copula constructions). In particular, both the contributing

variables $Y$ and predictors $X$ are time dependent, i.e. $\boldsymbol{Y} = \boldsymbol{Y}(t)$ and $\boldsymbol{X} = \boldsymbol{X}(t)$.

A particular type of such a model is obtained when the predictors are not considered in the joint pdf, i.e., when considering $f_{\boldsymbol{Y}}(\boldsymbol{Y})$. This *unconditional model* does not allow for changes of the contributing variables $Y$ and of the impact due to variations of the predictors $X$. In general, formalizing the impact $h$ of a CE as in step 1 - to then asses the risk of CE based on values of $h$ - corresponds to the *Structural Approach* (Salvadori et al., 2015; Serinaldi, 2015; Volpi and Fiori, 2014), which has recently

been formalized in Salvadori et al. (2016). Here, the advantage of the general model we propose is that it allows for taking into account variations of the impact $h$ driven by temporal changes of the predictors $X$. Through the conditional pdf, the model allows for a realistic representation both of the dependencies between the $Y_i$, and of their marginal distributions.

When the variables $Y$ are available but not the impact $h$, the model can still be used to only estimate the variables $Y$. This may be useful when assessing the risk of CEs through, e.g., multivariate return periods of the contributing variables $Y$ (e.g.

Graeler et al., 2016, 2013; Salvadori et al., 2016, 2011; Wahl et al., 2015; Aghakouchak et al., 2014; Saghafian and Mehdikhani, 2013; Shiau et al., 2007; Shiau, 2003). Moreover, it may happen that the impact $h$ is available, but the variables $Y$ are not. In this case the model may still be used in the form $f_{h|\boldsymbol{X}}(h|\boldsymbol{X})$ to directly estimate the impact $h$, based on the conditional joint pdf of the impact $h$, given the predictors $X$. In this case, depending on the physical system, it may be more or less complicated to calibrate the predictors. Also, we observe that equation (1) is general and a possibility for estimating the impact would be

to use the conditional joint pdf $f_{h|\boldsymbol{Y}}(h|\boldsymbol{Y})$. Such an approach may be useful for cases where complex relations exist between the impact $h$ and the variables $Y$, and therefore it may be difficult to implement, e.g., a proper regression model to describe the impact $h$.

An advantage of using a parametric statistical model is that this constrains the dependencies between the contributing variables, as well as their marginal distributions, and thereby reduces their uncertainties with respect to empirical estimates (Hobæk

Haff et al., 2015). Such a reduction in turn reduces the uncertainty in the estimated physical quantity of interest, like the impact of the CE. However, the uncertainty reduction depends on the choice of a proper parametric model, in particular when modelling the tail of a univariate or multivariate distribution.

## 4  Statistical method

Pair-copula constructions (PCCs) are mathematical decompositions of multivariate pdfs proposed by Joe (1996), which allow for the modelling of multivariate dependencies with high flexibility. We start presenting the concept of copulas, and then we introduce PCCs. More technical details can be found in the appendices.

### 4.1  Copulas

Consider a vector $\boldsymbol{Y} = (Y_1, ..., Y_n)$ of random variables, with marginal pdfs $f_1(y_1), ..., f_n(y_n)$, and cumulative marginal distribution functions (CDFs) $F_1(y_1), ..., F_n(y_n)$, defined on $\mathbb{R} \cup \{-\infty, \infty\}$. We use the recurring definition $u_i := F_i(y_i)$, where the name $u$ indicates that these variables are uniformly distributed by construction. According to Sklar's theorem (Sklar, 1959) the joint CDF $F(y_1, ..., y_n)$, can be written as:

$$F(y_1, ..., y_n) = C(u_1, ..., u_n) \tag{2}$$

where C is an n-dimensional Copula. C is a copula if $C : [0,1]^n \to [0,1]$ is a joint CDF of an n-dimensional random vector on the unit cube $[0,1]^n$ with uniform marginals (Joe, 2014; Salvadori et al., 2007; Nelsen, 2006; Genest et al., 2007; Salvadori and De Michele, 2007).

Under the assumption that the marginal distributions $F_i$ are continuous, the copula C is unique and the multivariate pdf can be decomposed as:

$$f(y_1, ..., y_n) = f_1(y_1) \cdot ... \cdot f_n(y_n) \cdot c(u_1, ..., u_n) \tag{3}$$

where $c$ is the copula density. Equation (3) explicitly represents the decomposition of the pdf as a product of the marginal distributions and the copula density, which describes the dependence among the variables independently of their marginals. Equation (3) has some important practical consequences: it allows us to generate a large number of joint pdfs. In fact, inserting any existing family for the marginal pdfs and copula density into eq. (3), it is possible to construct a valid joint pdf, provided that suitable constraints are satisfied. The group of the existing parametric families of multivariate distributions (e.g. the multivariate normal distribution, which has normal marginals and copula) is only a part of the realizations which are possible via equation (3). Copulas therefore make it easy to construct a wide range of multivariate parametric distributions.

### 4.2  Tail dependence

The dependence of extreme events cannot be measured by overall correlation coefficients such as the Pearson, Spearman or Kendall. Given two random variables which are uncorrelated according to such overall dependence coefficients, there can be a significant probability to get concurrent extremes of both variables, i.e., a tail dependence (Hobæk Haff et al., 2015). On the contrary, two random variables which are correlated according to an overall dependence coefficient may not necessarily be tail dependent.

Mathematically, given two random variables $Y_1$ and $Y_2$ with marginal CDFs $F_1$ and $F_2$ respectively, they are *upper tail dependent* if the following limit exists and is non-zero:

$$\lambda_U(Y_1, Y_2) = \lim_{u \to 1} P(Y_2 > F_2^{-1}(u)|Y_1 > F_1^{-1}(u)) \tag{4}$$

where $P(A|B)$ indicates the generic conditional probability of occurrence of the event $A$ given the event $B$. Similarly, the two variables are *lower tail dependent* if:

$$\lambda_L(Y_1, Y_2) = \lim_{u \to 0} P(Y_2 < F_2^{-1}(u)|Y_1 < F_1^{-1}(u)) \tag{5}$$

exists and is non-zero.

### 4.3 Pair-Copula Constructions (PCCs)

While the number of bivariate copula families is very large (Joe, 2014; Nelsen, 2006), building higher-dimensional copulas is generally recognised as a difficult problem (Aas et al., 2009). As a consequence, the set of copula families having dimension greater than or equal to 3 is rather limited, and they lack flexibility in modelling multivariate pdfs where heterogeneous dependencies exist among different pairs. For instance, they usually prescribe that all the pairs have the same type of dependence, e.g. they are either all tail dependent or all not tail dependent. Under the assumption that the joint CDF is absolutely continuous, with strictly increasing marginal CDFs, PCCs allow to mathematically decompose an n-dimensional copula density into the product of $n(n-1)/2$ bivariate copulas, some of which are conditional. In practice, this provides high flexibility in building high-dimensional copulas. PCCs allow for the independent selection of the pair-copulas among the large set of families, providing higher flexibility in building high dimensional joint pdfs with respect to using the existing multivariate parametric copulas (Aas et al., 2009).

When the dimension of the pdf is large, there can be many possible, mathematically equally valid decompositions of the copula density into a PCC. For example, for a 5 dimensional system there are 480 possible different decompositions. For this reason, Bedford and Cooke (2001b, 2002) have introduced *the regular vine*, a graphical model which helps to organize the possible decompositions. This is helpful to choose which PCC to use to decompose the multivariate copula. In this study we concentrate on the subcategories *canonical* (also known as *C-vine*) and *D-vine* of regular vines. Out of the 480 possible decompositions for a 5-dimensional copula density, 240 are regular vines (60 C-vines, 60 D-vines and 120 other types of vines) (Aas et al., 2009). The decomposition we selected for the conditional model is the following D-vine:

$$
\begin{aligned}
f_{12345}(y_1, y_2, y_3, y_4, y_5) = {} & f_4(y_4) \cdot f_5(y_5) \cdot f_3(y_3) \cdot f_1(y_1) \cdot f_2(y_2) \\
& \cdot c_{45}(u_4, u_5) \cdot c_{53}(u_5, u_3) \cdot c_{31}(u_3, u_1) \cdot c_{12}(u_1, u_2) \\
& \cdot c_{43|5}(u_{4|5}, u_{3|5}) \cdot c_{51|3}(u_{5|3}, u_{1|3}) \cdot c_{32|1}(u_{3|1}, u_{2|1}) \\
& \cdot c_{41|35}(u_{4|53}, u_{1|53}) \cdot c_{52|13}(u_{5|31}, u_{2|31}) \\
& \cdot c_{42|135}(u_{4|513}, u_{2|513})
\end{aligned}
\tag{6}
$$

where $(Y_1, Y_2, Y_3)$ are the variables $(Y_{1_{\text{Sea}}}, Y_{2_{\text{River}}}, Y_{3_{\text{River}}})$, and $(Y_4, Y_5)$ are the predictors $(X_{1_{\text{Sea}}}, X_{23_{\text{Rivers}}})$ (details about the predictors are given in the next section). Details about the selection procedure of the vine (eq. (6)) are given in appendices B2 and C, while the graphical representation of this vine is shown in Figure 10A (appendix B1).

As described in section 3, the conditional model is based on the conditional joint pdf $f_{\boldsymbol{Y}|\boldsymbol{X}}(\boldsymbol{Y}|\boldsymbol{X})$, which is decomposed via PCC. Details regarding conditional joint pdfs decomposed as C- or D-vines (including the developed algorithms for sampling from such vines) are presented in appendix B2. Moreover, the developed routines for working with conditional vines are publicly available via the R-package *CDVineCopulaConditional* (Bevacqua, 2017). More details about vines and the decompositions used for the unconditional model are given in appendix B1. Details regarding the statistical inference of the joint pdf can be found in appendix C.

## 5  Model development

The extreme impact of compound events may be driven from the joint occurrence of non-extreme contributing variables (Leonard et al., 2014; Seneviratne et al., 2012). This is the case for compound floods in Ravenna, where not all extreme values of the impact would be considered if selecting only extreme values of the contributing variables. Therefore we model the contributing variables, without focusing only on their extreme values. Below we show the steps we follow to study compound floods in Ravenna, based on the conceptual model described in section 3. We will go through these steps in detail in the next sections.

1. Define the impact function:

$$h = h(Y_{1_{\text{Sea}}}, Y_{2_{\text{River}}}, Y_{3_{\text{River}}}). \tag{7}$$

   The contributing variables $Y$ (sea and river levels) and the impact are shown in the black rectangle of Figure 2).

2. Find the meteorological predictors of the contributing variables $Y$. For each variable $Y_i$ we found more than one meteorological predictor, which we aggregated into a single variable $X_i$. We refer to this variable as the predictor $X_i$ of the variable $Y_i$ from now on. Moreover we use the same predictor for the two river levels because they are driven by a similar meteorological influence. The predictors are graphically shown in Figure 2, where we introduce $X_{1_{\text{Sea}}}$ (the predictor of $Y_{1_{\text{Sea}}}$) and $X_{23_{\text{Rivers}}}$ (the predictor of $Y_{2_{\text{River}}}$ and $Y_{3_{\text{River}}}$).

3. Fit the 5-dimensional conditional joint pdf $f_{\boldsymbol{Y}|\boldsymbol{X}}(Y_{1_{\text{Sea}}}, Y_{2_{\text{River}}}, Y_{3_{\text{River}}}|X_{1_{\text{Sea}}}, X_{23_{\text{Rivers}}})$ of the conditional model (modelled via PCC). To develop the unconditional model, we fit the 3-dimensional pdf $f_{\boldsymbol{Y}}(Y_{1_{\text{Sea}}}, Y_{2_{\text{River}}}, Y_{3_{\text{River}}})$, which includes only the contributing variables $Y$ inside the black rectangle of Figure 2. The time series of the contributing variables have significant serial correlations, and this should be considered in order to avoid underestimating the risk uncertainties (see appendix E and Figure 12). Only for the unconditional model, we explicitly modelled such serial correlations through combining the PCC with autoregressive $AR(1)$ models (see appendix E).

4. Given the complexity of the problem, an analytical derivation of the statistical proprieties of the impact is impracticable. Therefore, we apply a Monte Carlo procedure. Specifically we simulate the contributing variables $Y$ from the fitted models, and then we define the simulated values of $h$ via equation (7) as:

$$h^{\text{sim}} := h(Y_{1_{\text{Sea}}}^{\text{sim}}, Y_{2_{\text{River}}}^{\text{sim}}, Y_{3_{\text{River}}}^{\text{sim}}) \tag{8}$$

where $\boldsymbol{Y}^{\text{sim}}$ are the simulated values of $\boldsymbol{Y}$.

5. Perform a statistical analysis of the values $h^{\text{sim}}$. To asses the risk associated with the events, we compute the return levels of $h$ through fitting a Generalized Extreme Value (GEV) distribution to annual maximum values (defined over the period November-March). We compute the model uncertainties, which is straightforward through such models. Practically, such uncertainties propagate through to the risk assessment, and so they must be considered (details about model based return level uncertainty are given in appendix D).

To neglect the Monte Carlo uncertainties, i.e., the sampling uncertainties due to the model simulations, we produce long simulations. For example, to obtain the model based return level curve, we simulate a time series $h^{\text{sim}}(t)$ of length equal to 200 times the length of the observed data (6 years). From this we get a time series of 1200 annual maximum values, to whom we fit the GEV distribution to get the return level. Observation based return levels are obtained through fitting a GEV to annual maximum values of $h^{\text{obs}}$. The relative uncertainties are computed through propagating the parameter uncertainties of the fitted GEV distribution (more details are given at the end of appendix D).

## 5.1 Impact function

The water level $h$ is influenced by river ($Y_{2_{\text{River}}}$ and $Y_{3_{\text{River}}}$) and sea ($Y_{1_{\text{Sea}}}$) levels (Figure 2). We describe this influence through the following multiple regression model:

$$h = a_1 Y_{1_{\text{Sea}}} + a_{21} Y_{2_{\text{River}}} + a_{22} Y_{2_{\text{River}}}^2 + a_{31} Y_{3_{\text{River}}} + a_{32} Y_{3_{\text{River}}}^2 + c + \eta_{\text{h}}(0, \sigma_h) \tag{9}$$

where $\eta_{\text{h}}(0, \sigma_h)$ is a Gaussian distributed noise having standard deviation equal to $\sigma_h$. The contribution of the rivers to the impact $h$ is expressed via quadratic polynomials, which guarantees a better fit of the model according to the Akaike Information Criterion (AIC). In particular, we defined the regression model as the best output of both a forward and a backward selection procedure, considering linear and quadratic terms for all of the $Y$ as candidate variables. The Q-Q plot of the model, i.e. the plot of the quantiles of observed values against those of the mean predicted values from the model, is shown in Figure 3. The points are located along the line $y = x$, which indicates that the model is satisfying. Omitting one of the variables as predictor reduces model performance, underlining the compound nature of the impact $h$. The sum of the relative contributions of the rivers is very similar to that of the sea. The parameters of this model (and of those in section 5.2) were estimated according to the maximum likelihood approach, solved through QR decomposition (via the *lm* function of the R package *stats* (R Core Team, 2016)).

[Figure 3 about here.]

## 5.2 Meteorological Predictor Selection

Figure 4 shows the resulting scatter plots of observed predictands ($Y^{\text{obs}}$) and selected observed predictors ($X^{\text{obs}}$). To fit the joint pdf of the conditional model, we use all time steps where data for all of the $X$ and $Y$ variables have been recorded. However, we calibrate the predictors of rivers and sea separately, so we use all available data for each $Y$ variable (during the period November-March). The procedure we use to identify the meteorological predictors is shown below.

[Figure 4 about here.]

### 5.2.1 River levels

The meteorological influence on the two rivers $Y_{2_{\text{River}}}$ and $Y_{3_{\text{River}}}$ is very similar because their catchments are small and close by (as a consequence the Spearman correlation between the rivers is high, i.e. $0.79$). Therefore we use the same predictor for the two river levels.

The river levels are influenced by the total input of water over the catchments, which is given by the positive contribution of precipitation and snow melt, and by evaporation which results in a reduction of the river runoff. Specifically, we compute the input of water $w$ on the day $t^*$ over the river catchments (one grid point) as:

$$w(t^*) = P_{\text{total}}(t^*) - E(t^*) + S_{\text{melt}}(t^*) - S_{\text{fall}}(t^*) \tag{10}$$

where $P_{\text{total}}$ is the total precipitation, $E$ is the evaporation, $S_{\text{melt}}$ is the snow melt and $S_{\text{fall}}$ is the snow fall. The snow fall accounts for the fraction of precipitation which does not immediately contribute to the input of water over the catchments because of its solid state. While a fraction of the water input over the catchment rapidly reaches the rivers as surface runoff, another fraction infiltrates the ground and contributes only later to the river discharge. Compared with the first fraction, the second has a slower response to precipitation and changes more gradually over time. This double effect underlines the compound nature of river runoff whose response to precipitation falling at given time is higher if in the previous period additional precipitation fell in the river catchment. To consider both of these effects we define the river predictor as:

$$X_{23_{\text{Rivers}}}(t) = a_{\text{R}} \sum_{t^*=t-1}^{t} w(t^*) + b_{\text{R}} \sum_{t^*=t-10}^{t} w(t^*) + c_{\text{R}} \tag{11}$$

where $c_{\text{R}}$ is a constant. We choose the parameters of equation (11) through fitting the right hand side of this equation to the river contributions to the impact, i.e. $Y_{23_{\text{Rivers}}} := a_{21}Y_{2_{\text{River}}} + a_{22}Y_{2_{\text{River}}}^2 + a_{31}Y_{3_{\text{River}}} + a_{32}Y_{3_{\text{River}}}^2$ (see eq. (9)). The lags $n = 1$ and $n = 10$ days are those which maximise respectively the upper tail dependence and the Spearman correlation between $Y_{23_{\text{Rivers}}}(t)$ and the cumulated $w$ over the previous n days, i.e., $\sum_{t^*=t-n}^{t} w(t^*)$. Here, we use the upper tail dependence to get the typical river response time to the fraction of water which directly flows into the rivers as surface runoff. Similarly, the Spearman correlation is used to get the typical time required for the infiltrated water in the ground to flow into the rivers.

Through defining the river predictor as in equation (11), we aggregate the different meteorological drivers of the rivers in the single predictor $X_{23_{\text{Rivers}}}(t)$. Such aggregation allows for a simplification of the system describing the compound floods, due to

a reduction of the involved variables. Furthermore this reduces the variables described by the joint pdf $f_{\boldsymbol{Y},\boldsymbol{X}}(\boldsymbol{Y},\boldsymbol{X})$, whose numerical implementation errors can potentially increase with higher dimensionality (Hobæk Haff, 2012).

All of the terms involved in the multiple regression model (equation (11)) are statistically significant at level $\alpha = 2 \cdot 10^{-16}$. Moreover, the quality of the river predictor $X_{23_{\mathrm{Rivers}}}$ improves (according to the likelihood and to Spearman correlation between

$X_{23_{\mathrm{Rivers}}}$ and $Y_{23_{\mathrm{Rivers}}}$) when we use all of the terms in equation (10), instead of only $P_{\mathrm{total}}(t^*)$. The presence of more terms in equation (10) does not increase the number of model parameters.

### 5.2.2   Sea level

Sea level can be modeled as the superposition of the barometric pressure effect, i.e., the pressure exerted by the atmospheric weight on the water, the wind-induced surge, and an overall annual cycle. As for the river predictor, we aggregate the different

physical contributions in a single predictor. We define the sea level predictor on day $t$ as:

$$X_{1_{\mathrm{Sea}}}(t) = a_{\mathrm{S}}\, SLP_{Ravenna}(t) + b_{\mathrm{S}}\, \boldsymbol{SLP}(t) \cdot \boldsymbol{R}_{\mathrm{MAP}} + c_{\mathrm{S}}\, sin(\omega_{1\mathrm{Year}}t + \phi) + d_{\mathrm{S}} \tag{12}$$

where $SLP_{Ravenna}$ is the sea level pressure in Ravenna, $\boldsymbol{SLP} \cdot \boldsymbol{R}_{\mathrm{MAP}}$ is the wind contribution due to the sea level pressure field $SLP$, the harmonic term is the annual cycle and $d_{\mathrm{S}}$ is a constant term. We choose the parameters of equation (12) through regressing the sea level $Y_{1_{\mathrm{Sea}}}(t)$ on the right hand side of this equation. A more detailed physical interpretation of the terms is

given in the following.

1. $a_{\mathrm{S}} SLP_{Ravenna}$ accounts for the barometric pressure effect (Van Den Brink et al., 2004). The regression map $\boldsymbol{R}_{\mathrm{MAP}}$ indicates which anomalies of the SLP field are associated with high values of the residual of the barometric pressure effect (see Figure 5, where also more details are given). Particularly, according to the geostrophic equation for wind, these pressure anomalies induce wind in the Adriatic Sea towards Ravenna's coast. Therefore, the projection of the SLP

field onto this regression map, i.e, the term $\boldsymbol{SLP}(t) \cdot \boldsymbol{R}_{\mathrm{MAP}}$, describes the wind-induced change in sea level at time $t$.

2. $c_{\mathrm{S}}\ sin(\omega_{1\mathrm{Year}}t + \phi)$ describes the remaining annual cycle of the sea level which is not described by barometric pressure effect and wind contribution. This harmonic term could be driven by the annual hydrological cycle (Tsimplis and Woodworth, 1994), i.e., due to cyclic runoff of rivers which flow into the Adriatic sea, or due to density variations of the sea water (caused by the annual cycle of water temperatures). Astronomical tide may explain a minor fraction of this term.

The range of variation of $c_{\mathrm{S}}\ sin(\omega_{1\mathrm{Year}}t + \phi)$ is about $10\%$ of that of the sea level. When we use the predictor to extend the analysis to the period 1979-2015 this term will be kept constant assuming that the annual cycle has not drastically changed in past years. Moreover, we will not consider long-term sea level rise because its influence on both sea and impact $h$ level variations is negligible over the considered period (the observed rate of sea level rise in the North Adriatic Sea has been $\sim 0.8mm/year$ (NOAA, Tides & Currents)). Also the relative sea level rise has been negligible over the

considered period (Carbognin et al., 2011).

[Figure 5 about here.]

All the terms involved in the multiple regression model are statistically significant at level $\alpha = 2 \cdot 10^{-16}$.

## 6 Results

The results of the unconditional and conditional models are presented in the following sections.

### 6.1 Unconditional (3-dimensional) model

The unconditional model reproduces the joint pdf of the contributing variables $(Y_{1_\text{Sea}}, Y_{2_\text{River}}, Y_{3_\text{River}})$, and, in conjunction with the autoregressive models, also the serial correlations. The model is used to simulate values of the impact $h$ and assess the risk of compound floods, with related uncertainties. The selected pair-copula constructions and fitted pair-copula families are shown in appendices B1 and C.

Figure 6 shows, qualitatively, a good agreement between simulated and observed contributing variables $Y$.

[Figure 6 about here.]

In Figure 7 we show the return levels of the impact $h$. There is good agreement between the model and observation based expected return levels, even for return periods larger than six years (the length of the observed data). For return periods larger than shown in Figure 7, the agreement slowly decreases. The model based expected return period of the highest compound flood observed (3.19 m) is 18 years (the 95% confidence interval is $[2.5, \infty]$ years, where $\infty$ indicates a value larger than $10^{50}$ in this context from now on). The reason for such large uncertainty in the return period is the shortness of available data. However, the model based uncertainties are large but still smaller, up to return periods of about 60 years, than those obtained when computing the return level directly (based on the GEV) on the observed data of the impact (Figure 7). Moreover, when considering a model which does not take the serial correlation of the contributing variables $Y$ into account, we get an underestimation of the risk uncertainties. For example, the amplitude of the 95% confidence interval of the 20-years return level is underestimated by about 50% (not shown).

[Figure 7 about here.]

### 6.2 Conditional (5-dimensional) model

This model allows for assessing the change in the risk of compound floods due to temporal variations of the meteorological predictors of the contributing variables $Y$. We calibrate the model to the period 2009-2015. After validating the model for the period 2009-2015, we use predictors of the period 1979-2015 to extend the analysis of compound flood risk to the past. The selected pair-copula construction and fitted pair-copula families are shown in appendices B1 and C. We assess the quality of the model comparing predictions with observations. Specifically we look at its overall accuracy through considering the root-mean-square error between model predictions and observed data. Moreover we look at the accuracy of the model when predicting extreme values of the impact $h$ (defined as values of $h$ larger than the 95-percentile of $h^\text{obs}$), using the Brier score (see appendix F). To assess the quality of the model, avoiding overfitting, we perform a 6-fold cross-validation (see appendix G).

[Figure 8 about here.]

The cross-validation time series of the impact $h$ is visually compared with $h^{\text{obs}}$ in Figure 8. The average of the simulated cross-validation time series in general follows the temporal progression of $h^{\text{obs}}$ (Figure 8), and about $94\%$ of the observed impact values lie within the $95\%$ prediction interval. In particular, the highest flood observed is well predicted and lies inside the prediction interval. The Brier score based on the cross-validation time series is $BS_{CV} = 0.029$, while that relative to the reference model, i.e. the climatology (see appendix F), is $BS_{CL} = 0.046$. The resulting Brier skill score is $BSS = 1 - \frac{BS_{CV}}{BS_{CL}} = 0.38$, which indicates that the model is more accurate than the reference model in predicting extreme values of the impact $h$. In general, the skill of the model, both in terms of root-mean-square error and Brier score, does not change much when the cross-validation is not performed. This underlines that no artificial skill is present in the model. These positive results provide good confidence for extending the impact time series to the period 1979-2015. It also makes the model potentially interesting for flood forecasting and warning.

In Figure 9A we show the return levels of the impact $h$. As in the unconditional model, return levels are stationary, i.e., estimated through fitting a stationary GEV distribution to annual maximum values. The discrepancy between model and observation based return levels for the conditional model is smaller than for the unconditional, in particular for high return periods. It may happen that the dependencies between river and sea levels are not considered in some analyses when assessing the risk of flooding. Kew et al. (2013) show in Rotterdam, which is affected by floods driven both from surge and river discharges, that the boundary conditions used to build the protection barrier were determined assuming independence between sea level and river discharge. Here we observe that ignoring such a dependence may result in an underestimation of the estimated risk. The expected return period of the highest compound flood observed (3.19 m), computed over the period 2009-2015, is 20 years (the $95\%$ confidence interval is $[4.9, \infty]$ years). When not considering the dependencies between river and sea levels, the expected return period of the highest compound flood observed increases to 32 years (the $95\%$ confidence interval is $[6.7, \infty]$ years). Figure 9B shows that the return level estimates are reduced by about $0.2$ m when not considering such dependencies between sea and river levels. In particular, at the $95\%$ confidence level, the return levels are underestimated when not considering these dependencies for return periods smaller than about 40 years. The same, however, cannot be clearly concluded for return periods larger than 40 years because of the large uncertainties (Figure 9B). A similar result is obtained from the unconditional model (not shown). Therefore, although there is not a large difference in the return levels when treating sea and rivers independently or not, in Ravenna it may be relevant to incorporate their dependencies into the flood risk estimation. An imprecise risk assessment may bring negative societal consequences due to inadequate information provided for infrastructural adaptation.

To estimate the risk based on predicted values of the impact during the past, we run the simulations through conditioning on predictors of the period 1979-2015. This allows us to get a more robust estimation of the risk compared to that obtained considering only the period 2009-2015. The return levels in Figure 9A (dashed line), are similar to that estimated when analysing the period 2009-2015. Although this result suggests a stationarity of the risk during the period 1979-2015, we investigate if there has been any trend in the risk during the recent past. To do this, we computed time dependent return levels. Specifically, we computed stationary return levels on moving temporal windows of six years during the period 1979-2015, based on $h^{\text{sim}}$ values obtained through conditioning on predictors belonging to these windows. However, we did not observe any long-term

trend in the risk. Moreover, analysing the return levels computed on moving temporal windows during the period 1979-2015, we did not observe any long-term trend neither in the risk of storm surge nor in that of river floods (not shown).

During the period 1979-2015, there has not been a long-term trend in the risk due to a variation of the marginal distributions of the predictors, or in their dependence. To study this, we computed the return levels on moving temporal windows in the cases described below. First, we simulated the impact through conditioning the $Y^{\mathrm{sim}}$ variables on predictors having the observed marginal distributions of the period 1979-2015, but fixing the dependence to that observed during 2009-2015. Secondly, we simulated the impact through conditioning on predictors having the observed dependence of the period 1979-2015, and fixed marginal distributions to the ones observed during 2009-2015. In both cases we did not find any long-term trend in the return levels (not shown).

[Figure 9 about here.]

## 7 Discussion and Conclusions

Compound events (CEs) are multivariate extreme events in which the contributing variables may not be extreme themselves, but their joint - dependent - occurence causes an extreme impact. Conventional univariate statistical analysis cannot give accurate information regarding the multivariate nature of CEs, and therefore on the risk associated with these events.

We develop a conceptual model, implemented via pair-copula constructions (PCCs), to quantify the risk of CEs, as well as the associated sampling uncertainty. This model includes predictors, which could represent for instance meteorological processes. The inclusion of predictors in the model (1) provides insight into the physical processes underlying CEs, as well as into the temporal variability of CEs, and (2) allows to statistically downscale CEs and their impacts. The model is in principle extendable to any number of contributing variables and predictors, given a large enough sample of data for calibration.

Downscaling may be used to statistically extend the risk assessment back in time to periods where observations of the predictors are available, but not of the contributing variables and impacts, or to assess potential future changes in CEs based on climate models. The conceptual model is particularly useful to downscale large scale predictors from climate models in cases where the local contributing variables driving the impacts of CEs are either not realistically simulated, or not simulated at all by the available climate models. As such, the model can straightforwardly be used to assess future risk of CEs based on multi model ensembles as available from the CMIP (Taylor et al., 2012) and CORDEX (Giorgi et al., 2009) archives.

The model makes use of PCCs, a very powerful statistical method to model multivariate dependencies. PCCs are particularly useful to model CEs, when the contributing variable pairs have different dependence structures, e.g., when only some of them are characterised by tail dependence. To model such types of structures, even multivariate parametric copulas, which have been introduced in climate science to overcome some difficulties in modelling multivariate density distributions (e.g. Schölzel and Friederichs, 2008), lack of flexibility. PCCs are more convenient: through decomposing the dependence structure into bivariate copulas, they give high flexibility in modelling generic high dimensional systems. We suggest to consider the use of PCCs for modelling compound events which involve more than two contributing variables, or when predictors are included in the system as additional variables.

The model allows for a straightforward quantification of sampling uncertainties. In many cases, such risk uncertainties might be substantial as observed data are often limited, and should thus be quantified. In fact, uncertainty estimates are essential to avoid drawing conclusions that may be misleading when uncertainties are large (as also recently discussed by Serinaldi (2015)).

We adapt the developed conceptual model to study compound floods in Ravenna, which are floods driven by the joint occurrence of storm surge and high river level. Namely, the contributing variables of the compound floods are the river and sea levels, whose combination drives the impact, i.e., the water level in between the river and the sea.

We used the specific adaptation of the model to statistically downscale the river and sea level from meteorological predictors, and therefore estimate the impact of the compound floods as a function of the downscaled sea and river levels. The accuracy of the estimated impact appears satisfactory, such that the model is potentially interesting for use in both flood forecasting and warning. Also, the model based expected return levels of the impact are about the same as those directly computed on observed data of the impact. Although the model based uncertainty on these return levels is very large (due to the shortness of the available data), for return period smaller than about 60 years it is smaller than that obtained computing the risk directly on the observed data of the impact.

We calibrate the model over the period 2009-2015, and by including meteorological predictors obtained from the *ECMWF ERA-Interim* reanalysis dataset, we extend the analysis of compound flooding to the full period of 1979-2015, to obtain a more robust estimation of the risk. The expected return period of the highest compound flood observed, computed over the period 1979-2015, is 19 years (the $95\%$ confidence interval is $[3.7, \infty]$ years). Moreover, we did not observe any long-term trend in risk during the period 1979-2015.

Ignoring the estimated dependence between sea and river levels may lead to an underestimation of risk. Specifically, assuming independence between sea and river levels, the expected return period of the highest compound flood observed - computed over the period 2009-2015 - is 32 years (the $95\%$ confidence interval is $[6.7, \infty]$ years). When assuming the estimated dependence between sea and river levels, it decreases to 20 years (the $95\%$ confidence interval is $[4.9, \infty]$ years). In other cities affected by sea surges and river flooding, e.g., in Rotterdam, protection barriers were designed assuming independence between sea level and river discharge (Kew et al., 2013), a decision which is still debated about (Van Den Brink et al., 2005; Kew et al., 2013; Klerk et al., 2015). In Ravenna, it may be relevant to incorporate these dependencies into the flood risk estimation. An imprecise risk assessment may harm the population at risk due to inadequate information provided for infrastructural adaptation. In general, when considering generic CEs, their associated risk may be substantially influenced by the dependence between the contributing variables, and so this dependence should be considered.

In the context of compound floods, only a few studies have explicitly quantified the impact and the associated risks (Zheng et al., 2015, 2014; Van den Hurk et al., 2015). This might be due to the practical difficulties in quantifying the impact. For example, to quantify the impact of compound floods in the river mouth, it is necessary to have water level data at a station where both the influence of sea and river are seen. However, we have found few locations where these stations exist as, maybe in part, stakeholders are usually interested in data where only the influence of the river or the sea is seen. Also, for places where data show both the influence of sea and river, the measurements can be affected by human influences such as pumping stations between river and sea stations. Moreover, while compound floods involve a dependence between sea and river levels

(Leonard et al., 2014), places where there are stations detecting both the influence of sea and river may not present such dependence. Therefore, we argue that to obtain a more in-depth knowledge of these events, it may be very useful to create an archive containing data for locations where compound floods have been recorded, and eventually increase the effective number of measurements in places which are supposed to be under risk of compound floods.

## 8  Code availability

The developed routines for working with conditional joint probability density functions decomposed as D- or C-vines are publicly available via the R-package *CDVineCopulaConditional* (Bevacqua, 2017) (more details are given in appendix B2).

## 9  Data availability

Sea level data of the station Ravenna-Porto Corsini were downloaded from the *Italian National Institute for Environmental Protection and Research (ISPRA)*, and are available under the link: www.mareografico.it. River data can be downloaded from *Arpae Emilia-Romagna*, via the link www.arpae.it/dettaglio_generale.asp?id=3284&idlivello=1625 (the names of used stations are S. Marco, S. Bartolo and Rasponi, where the latter is that used for the impact).

## Appendix A:  Homogenisation of river level data

The *zero reference level* of river measurements is the water level in the river defined as zero in the measurements. In general, such a zero reference level may change during different periods of observation, due to technical reasons. As the zero reference level of rivers $Y_{2_{\text{River}}}$ and $Y_{3_{\text{River}}}$ varied in the first three years but remained constant in the second three, we homogenised the former with respect to the latter at both rivers. We performed such homogenisation assuming that the precipitation falling into the catchment during one year is responsible for the average river level in the same year. For each river $Y_{i_{\text{River}}}$, we fitted the linear model $Y_{i_{\text{River}}}^{\text{annual}} = a_i P_i^{\text{annual}} + b_i$ in the last three years (those having constant zero reference level), where $Y_{i_{\text{River}}}^{\text{annual}}$ is the annual average of $Y_{i_{\text{River}}}$ and $P_i^{\text{annual}}$ is the annual cumulated precipitation over the river basin (data from *ECMWF ERA-Interim* reanalysis dataset). Finally, for each river, we translated the zero reference level of the first three years, such that the linear model was valid in these years as well.

## Appendix B:  Vines and sampling procedure

In this appendix we show more details about vines, focusing on C- and D-vines. Moreover we discuss the sampling procedure, showing the algorithms to perform the conditional sampling from a C- and D-Vine.

## B1 Vines

Shown below are the general expressions to decompose an n-dimensional pdf via a PCC as C-vine (eq. (B2)) or D-vine (eq. (B1)) (Aas et al., 2009):

$$f_{Y_1,...,Y_n}(y_1,..,y_n) = \prod_{k=1}^{n} f(y_k) \prod_{j=1}^{n-1} \prod_{i=1}^{n-j} c_{i,i+j|i+1,...,i+j-1}\{F(y_i|y_{i+1},...,y_{i+j-1}), F(y_{i+j}|y_{i+1},...,y_{i+j-1})\} \tag{B1}$$

$$f_{Y_1,...,Y_n}(y_1,..,y_n) = \prod_{k=1}^{n} f(y_k) \prod_{j=1}^{n-1} \prod_{i=1}^{n-j} c_{j,j+i|1,...,j-1}\{F(y_j|y_1,...,y_{j-1}), F(y_{j+i}|y_1,...,y_{j-1})\}. \tag{B2}$$

The 5-dimensional vine that we use for the conditional model is shown in equation (6). The graphical representation of that decomposition is shown in Figure 10A, where the concept of *tree* is introduced. We show below the vines that we use for the unconditional model.

### B1.1 3-Dimensional vine

In total, a 3-dimensional copula density can be decomposed in three different ways, and each of these vines is both a D-vine and a C-vine. For this application we use the following vine.

$$\begin{aligned} f_{123}(y_1,y_2,y_3) = {} & f_1(y_1) \cdot f_2(y_2) \cdot f_3(y_3) \\ & \cdot c_{12}(u_1,u_2) \cdot c_{23}(u_2,u_3) \\ & \cdot c_{13|2}(u_{1|2},u_{3|2}). \end{aligned} \tag{B3}$$

This decomposition is represented graphically in Figure 10B. We underline that, in equation (B3), the rigorous expression of the conditional copula density $c_{13|2}$, of the pair $(U_1,U_3)$ given $U_2 = u_2$, would be $c_{13|2}(u_{1|2},u_{3|2};u_2)$. In equation (B3), $c_{13|2}$
is written under the assumption of a *simplified PCC*, i.e. the parameters of $c_{13|2}$ are the same for all values of $u_2 \in (0,1)$. The simplified PCC may be a rather good approximation, even when the simplifying assumption is far from being fulfilled by the actual model (Hobæk Haff et al., 2010; Stöber et al., 2013). Copula parameters that are functions of the conditioning variables, and thus violate the simplifying assumption, are approximated by the average over all values of the conditioning variables. The effect of this approximation on the estimated impact is likely to be small (Hobæk Haff et al., 2010; Stöber et al., 2013).

In this study of compound floods, the variables $(Y_1,Y_2,Y_3)$ of equation (B3) are the $(\varepsilon_{1_{\text{Sea}}}, \varepsilon_{2_{\text{River}}}, \varepsilon_{3_{\text{River}}})$ introduced in appendix E. Specifically, the vine of equation (B3) represents that used at the first step of the procedure in appendix D. The vine that we use at the third step of the procedure in appendix D is:

$$\begin{aligned} f_{123}(y_1,y_2,y_3) = {} & f_3(y_3) \cdot f_1(y_1) \cdot f_2(y_2) \\ & \cdot c_{31}(u_3,u_1) \cdot c_{12}(u_1,u_2) \\ & \cdot c_{32|1}(u_{3|1},u_{2|1}) \end{aligned} \tag{B4}$$

where $(Y_1,Y_2,Y_3) = (Y_{1_{\text{Sea}}}, Y_{2_{\text{River}}}, Y_{3_{\text{River}}})$.

[Figure 10 about here.]

## B2 Sampling procedure

To simulate a vector $\boldsymbol{Y} = (Y_1, ..., Y_n)$ of random variables, with marginal CDFs $F_1(y_1), ..., F_n(y_n)$, whose joint pdf is modelled via a copula, we first simulate from the copula the uniform variables $U_i$ for $i = 1, ..., n$ ($u_i := F_i(y_i)$), and then transform them into $Y_i$ for $i = 1, ..., n$ ($y_i := F_i^{-1}(u_i)$).

### 5 B2.1 Sampling and conditional sampling from vines

The simulation of the uniform variables from vines is discussed in Bedford and Cooke (2001a, b) and Kurowicka and Cooke (2005). Aas et al. (2009) show the algorithms to sample uniform variables from C- and D-vines. Due to the nature of PCCs, the sampling procedure works as a cascade. Once the first variable is simulated from a uniform distribution, each following variable is simulated as conditioned on the previous group of simulated variables.

10    It is clear then, that to sample from the conditional distribution of $U_{N_{\mathrm{cond}}+1}, ..., U_n$ given values for $U_1, ..., U_{N_{\mathrm{cond}}}$ (i.e. $f_{U_{N_{\mathrm{cond}}+1}, ..., U_n | U_1, ..., U_{N_{\mathrm{cond}}}}$), it is possible to follow this procedure by simply fixing the first $N_{\mathrm{cond}}$ variables at the conditioning values. The approach used here to execute such a procedure, is to select vines from which the conditioning variables would be sampled as first when following the sampling algorithms from Aas et al. (2009). For example, using the D-vine represented in Figure 10A (or in eq. (6)), we could simulate by fixing the pairs $(U_4, U_5)$ or $(U_2, U_1)$ in case we are interested in conditioning 15    the simulation on two variables.

Following this approach, for D-vines the number of n-dimensional decompositions which allow for conditioning on $N_{\mathrm{cond}}$ variables is $N_{\mathrm{cond}}! \cdot (n - N_{\mathrm{cond}})!$. For C-vines the number of the decompositions which allow for such a conditioning is $N_{\mathrm{cond}}! \cdot (n - N_{\mathrm{cond}})!/2$ for $n - N_{\mathrm{cond}} > 1$, and $N_{\mathrm{cond}}!$ for $n - N_{\mathrm{cond}} = 1$. For example, in this study we model a 5-dimensional system with two conditioning variables (the meteorological predictors), that is $n = 5$ and $N_{\mathrm{cond}} = 2$. Considering that there are not 20    5-dimensional vines which belong to both the C-vine and D-vine categories (Aas et al., 2009), the choice of the vine used for the model is done among $(2!/2 \cdot (5-2)!) + (2! \cdot (5-2)!) = 18$ vines. Furthermore, we need to condition on values $(y_4, y_5)$, therefore we simulate from the copula through conditioning on $(u_4 = F_4(y_4), u_5 = F_5(y_5))$, where $F_4$ and $F_5$ are the fitted marginals in the calibration period, while $(y_4, y_5)$ could theoretically be any value.

To apply such a sampling procedure, we developed the Algorithms 1 and 2, which are a modified version of Algorithms 1 and 25    2 shown in Aas et al. (2009). The developed algorithms allow for conditional sampling from a C- or a D-vine from which the conditioning variables would be sampled as first when following the sampling algorithms from Aas et al. (2009). Specifically, given a C- or a D-vine of the variables $(X_1, ..., X_{N_{\mathrm{cond}}}, X_{N_{\mathrm{cond}}+1}, ..., X_n)$, Algorithms 1 and 2 allow for the conditional sampling of $(X_{N_{\mathrm{cond}}+1}, ..., X_n)$ given $(X_1 = x_1^{\mathrm{cond}}, ...., X_{N_{\mathrm{cond}}} = x_{N_{\mathrm{cond}}}^{\mathrm{cond}})$, where $N_{\mathrm{cond}}$ is the number of conditioning variables. When conditioning variables are not given ($N_{\mathrm{cond}} = 0$), Algorithms 1 and 2 reduce to the special cases of Algorithms 1 and 2 shown 30    in Aas et al. (2009). Both routines relative to Algorithms 1 and 2 are publicly available via the R-package *CDVineCopula-Conditional* (Bevacqua, 2017). *CDVineCopulaConditional* includes tools to select the best vine (based on information criteria) among those which allow for such conditional sampling, and therefore to fit the pair-copula families.

**Algorithm 1** Algorithm to simulate uniform variables $\boldsymbol{X} = (X_1, ..., X_{N_{\mathrm{cond}}}, X_{N_{\mathrm{cond}}+1}, ..., X_n)$ from a C-vine. Generates one sample $x_{N_{\mathrm{cond}}+1}, ..., x_n$ conditioned on given values $x_1^{\mathrm{cond}}, ...., x_{N_{\mathrm{cond}}}^{\mathrm{cond}}$. The $h$-function is defined as in Aas et al. (2009). $\Theta_{j,i}$ is the set of parameters of the copula density $c_{j,j+1|1,...,j-1}$.

---

Sample $w_{N_{\mathrm{cond}}+1}, ..., w_n$ independent uniform on [0,1].

**if** $N_{\mathrm{cond}} \neq 0$ **then**

    **for** $i$ $in$ $(1, ..., N_{\mathrm{cond}})$ **do**

        $w_i = x_i^{\mathrm{cond}}$

    **end for**

**end if**

$x_1 = v_{1,1} = w_1$

**for** $i$ $in$ $(2, ..., n)$ **do**

    $v_{i,1} = w_i$

    **if** $i > N_{\mathrm{cond}}$ **then**

        **for** $k$ $in$ $(i-1, i-2, ..., 1)$ **do**

            $v_{i,1} = h^{-1}(v_{i,1}, v_{k,k}, \Theta_{k,i-k})$

        **end for**

    **end if**

    $x_i = v_{i,1}$

    **if** $i == n$ **then**

        **Stop**

    **end if**

    **for** $j$ $in$ $(1, ..., i-1)$ **do**

        $v_{i,j+1} = h(v_{i,j}, v_{j,j}, \Theta_{j,i-j})$

    **end for**

**end for**

---

Finally, we underline that this is not the only way to proceed for the conditional simulation (Bedford and Cooke, 2001b), but despite the fact that the best vine is selected among a fraction of all the possible, it can provide very satisfying results, as we show in this study. Also, we refer to Brechmann et al. (2013) and Liu et al. (2015) as other works where conditional joint pdfs decomposed as C-vines were used for statistical modelling.

## 5   Appendix C: Statistical inference of the joint pdf

Statistical inference on a pdf decomposed via a PCC is in principle very computationally demanding. As can be seen from equation (B3), the arguments of the copulas are influenced from the choice of the marginals (because of $u_i = F_i(x_i)$), and the argument of the copula in each level, is influenced from the fit of the copulas in the previous levels too. As a consequence of

**Algorithm 2** Algorithm to simulate uniform variables $\boldsymbol{X} = (X_1, ..., X_{N_{\text{cond}}}, X_{N_{\text{cond}}+1}, ..., X_n)$ from a D-vine. Generates one sample $x_{N_{\text{cond}}+1}, ..., x_n$ conditioned on given values $x_1^{\text{cond}}, ...., x_{N_{\text{cond}}}^{\text{cond}}$. The $h$-function is defined as in Aas et al. (2009). $\Theta_{j,i}$ is the set of parameters of the copula density $c_{i,i+j|i+1,...,i+j-1}$.

---

Sample $w_{N_{\text{cond}}+1}, ..., w_n$ independent uniform on [0,1].

**if** $N_{\text{cond}} \neq 0$ **then**

    **for** $i$ $in$ $(1, ..., N_{\text{cond}})$ **do**

        $w_i = x_i^{\text{cond}}$

    **end for**

**end if**

$x_1 = v_{1,1} = w_1$

**if** $N_{\text{cond}} < 2$ **then**

    $x_2 = v_{2,1} = h^{-1}(w_2, v_{1,1}, \Theta_{1,1})$

**else**

    $x_2 = v_{2,1} = w_2$

**end if**

$v_{2,2} = h(v_{1,1}, v_{2,1}, \Theta_{1,1})$

**for** $i$ $in$ $(3, ..., n)$ **do**

    $v_{i,1} = w_i$

    **if** $i > N_{\text{cond}}$ **then**

        **for** $k$ $in$ $(i-1, i-2, ..., 2)$ **do**

            $v_{i,1} = h^{-1}(v_{i,1}, v_{i-1,2k-2}, \Theta_{k,i-k})$

        **end for**

        $v_{i,1} = h^{-1}(v_{i,1}, v_{i-1,1}, \Theta_{1,i-1})$

    **end if**

    $x_i = v_{i,1}$

    **if** $i == n$ **then**

        **Stop**

    **end if**

    $v_{i,2} = h(v_{i-1,1}, v_{i,1}, \Theta_{1,i-1})$

    $v_{i,3} = h(v_{i,1}, v_{i-1,1}, \Theta_{1,i-1})$

    **if** $i > 3$ **then**

        **for** $j$ $in$ $(2, ..., i-2)$ **do**

            $v_{i,2j} = h(v_{i-1,2j-2}, v_{i,2j-1}, \Theta_{j,i-j})$

            $v_{i,2j+1} = h(v_{i,2j-1}, v_{i-1,2j-2}, \Theta_{j,i-j})$

        **end for**

    **end if**

    $v_{i,2i-2} = h(v_{i-1,2i-4}, v_{i,2i-3}, \Theta_{i-1,1})$

**end for**

---

this, the estimation of the parameters of the full pdf (marginals and pair-copulas) should be performed together. Moreover the structure of the vine has to be chosen, increasing the demands of computational resources.

To overcome these obstacles, some techniques have been developed. The complications regarding the dependence of the copula parameters from the marginals estimation can be overcome using empirical marginals (Genest et al., 1995). This allows for the estimation of copula parameters without the need of considering the marginals. However, to take into account that the estimation of the parameters of each pair copula depends on those of the upper levels, the estimation of the parameters of all the pairs should be performed at the same time. This way of estimating the parameters is called semiparametric (SP). The estimator we use here is the stepwise semiparametric (SSP). It was proposed by Aas et al. (2009) and then Hobæk Haff (2013), and despite being asymptotically less efficient than the SP (Hobæk Haff, 2013), it produces very satisfactory results and speeds up the procedure considerably (Hobæk Haff, 2012). As in SP, the PCC parameters are estimated independently of the marginals, but the estimation of the PCC parameters is performed level by level, plugging in the parameters from previous levels at each step (Hobæk Haff, 2012).

In this study of compound floods, for each marginal pdf we use a mixture distribution composed of the empirical and the Generalized Pareto Distribution (GPD) for the extreme. For each predictor $X$, the GPD is fitted to data above a threshold defined here as their respective 95-percentile. For each of the contributing variables $Y$, this threshold was chosen requiring that the mean of the simulated extreme values from the joint pdf, was as near as possible to the maximum observed value of the variable $Y$ we were fitting. Adding the GPD to the empirical marginal for the extremes is necessary so to not constrain the model to simulate values of the variables $Y$ with maximum values that never exceed those observed during the calibration period.

We use the AIC to select the best vine structure among C- and D-vines (those selected are shown in sections B1.1 and 4.3). In particular, for every possible C- and D-vine, we fit all possible families through the maximum likelihood estimation, and then we select the best family according to the AIC. Then, we select the best vine according to the AIC for the full model. The pair-copula families are chosen among those available in the R package *VineCopula* (Schepsmeier et al., 2016). In particular, for the unconditional model all of the available families are considered during the selection, while for the conditional model we restricted the choice to the first 31 families listed in the documentation file of the package. This is because of technical issues regarding the simulation of data from the conditional pdf of the conditional model. Once the vine is selected, to better assess the quality of the fit of each pair-copula, we use the K-plot (Figure 11). This is a plot of the Kendall-function $K(w) = P(C_{i,j}(U_i, U, j) \leq w)$ computed with the fitted copula, against $K(w)$ computed with the empirical copula obtained from the observed uniform data. This diagnostic plot indicates a good quality of the fit when the points follow the diagonal (Genest et al., 2007; Hobæk Haff et al., 2015). We note that the $K(w)$ of the fitted copula is computed using Monte Carlo methods (long simulations allow for neglecting the associated sampling error). In Figure 11 we show the resulting K-plots and the selected copulas with their respective parameters for the 5-dimensional PCC (K-plots for the 3-dimensional are not shown). The families chosen for copulas $c_{43|5}(u_{4|5}, u_{3|5})$ and $c_{42|135}(u_{4|513}, u_{2|513})$ according to the AIC were describing slightly negative dependencies ($< 0.1$), but for physical reasons we expect these copulas to describe slightly positive dependencies. We argue that this result is due to uncertainties of the model. Therefore we choose independent copulas for these pairs, which

is a compromise between the expert knowledge we have about the data and the result of the fit. When assuming independent copulas for these two pairs, the corresponding K-plots show only a small deviation from the diagonal (right side of Figure 11). Moreover these K-plots are mostly inside the $95\%$ confidence interval of the K-plots, which confirms the reasonability of choosing these two independent copulas.

[Figure 11 about here.]

The R packages *CDVineCopulaConditional* (Bevacqua, 2017) and *VineCopula* (Schepsmeier et al., 2016) were used to work with copulas. The GPDs for the marginal distributions were fitted through the function *gpd.fit* of the R package *ismev* (Heffernan and Stephenson, 2016).

## C1    Selected pair-copula families

In the case of the unconditional model, the fitted pair-copula families to the observed contributing variables $Y$ - relative to the vine of equation (B4) - are: *Survival BB1* (parameters: 0.49, 1.15) for $c_{31}(u_3, u_1)$, *BB8* (parameters: 4.01, 0.6) for $c_{12}(u_1, u_2)$, *Tawn type 1* (parameters: 2.59, 0.73) for $c_{32|1}(u_{3|1}, u_{2|1})$. The selected families relative to the vine of equation (B3), i.e. the one fitted to $(\varepsilon_{1_{\text{Sea}}}, \varepsilon_{2_{\text{River}}}, \varepsilon_{3_{\text{River}}})$ introduced in appendix E, are: *t-copula* (parameters: 0.15, 3.44) for $c_{12}(u_1, u_2)$, *Tawn type 2* (parameters: 2.85, 0.71 ) for $c_{23}(u_2, u_3)$, *Survival Gumbel* (parameter: 1.13) for $c_{13|2}(u_{1|2}, u_{3|2})$. In the case of the conditional

model, the selected pair-copula families with relative parameters, fitted to the observed data of contributing variables $Y$ and predictors $X$, are shown in Figure 11.

## Appendix D:  Model and risk uncertainty estimation via parametric bootstrap

The flexibility of copula theory to model multivariate distributions has determined its spread in literature, and more recently in climate science. However, once the model is fitted to observed data, we stress that procedures to get an estimate of the

uncertainties, both in the parameter estimates and the choice of the model, should be considered. This is particularly important, as it often happens that because of the limited sample size of the available data, these uncertainties are large and so cannot be neglected (Serinaldi, 2015). Practically they have a direct influence on the uncertainties of risk analysis. In particular, we observed that the uncertainties are also controlled by the dependence values between the modelled pairs (not shown).

In this study, we find model uncertainties in the joint pdf which propagate into large uncertainties when assessing the risk

of compound floods. This does not mean that such models are not useful, but instead that the results should be interpreted being aware of these existing uncertainties. Also, even if large, the obtained uncertainties in the risk are smaller than those obtained computing the risk analysis directly on the observed data of the impact, underlining another advantage of applying such procedures.

Both for the unconditional and conditional model, we use a parametric bootstrap to assess the model and subsequent risk

uncertainty, as follows:

1. Select and fit a model that can reproduce the statistical characteristics of $\mathbf{Y}^{\text{obs}}$: dependence among the variables and their marginal distributions (for the unconditional model we include also the serial correlation as described in appendix E).

2. Simulate $B = 2.5 \cdot 10^3$ samples of the contributing variables $Y$ (as well as predictors $X$ for the conditional model) with the same length as the observed data.

3. On each of the $B = 2.5 \cdot 10^3$ samples, fit a joint pdf via PCCs (the structure of the PCC is the same as that fitted on the observed data, while the pair-copulas families are re-selected for each sample).

4. From each of these $B = 2.5 \cdot 10^3$ models, simulate a sample of contributing variables $Y$ of length equal to 200 times the observed (for the conditional model the contributing variables $Y$ are simulated as conditioned on the predictors $X$).

5. For each sample, compute the simulated impact sequence as $h^{\text{sim}} = h(Y_{1_{\text{Sea}}}^{\text{sim}}, Y_{2_{\text{River}}}^{\text{sim}}, Y_{3_{\text{River}}}^{\text{sim}})$ and estimate the corresponding
10
return level curves. Return levels are estimated through fitting the Generalized Extreme Value (GEV) distribution on annual maximum values. We simulated samples of length 200 times the length of the observed data (6 years), to get - for each sample - 1200 annual maximum values on which we fit the GEV distribution. This allows us to neglect the uncertainty of the return levels driven by the sampling because the uncertainties of the GEV distribution parameters are negligible.

6. Estimate the uncertainties on the return levels through identifying the $95\%$ confidence interval (i.e. the range $2.5-97.5\%$) of the $B = 2.5 \cdot 10^3$ return level curves.

As underlined in step 1, for the unconditional model, we explicitly model the serial correlations of the contributing variables $Y$ when computing the uncertainties. This was done to avoid an underestimation of the risk uncertainties (see appendix E). For the conditional model, step 3 is a rigorous bootstrap procedure, while for the unconditional model this step is an approximation.
In fact, for the unconditional model, at step 3 we should have fitted the same type of model as fitted in step 1, i.e. that could include the serial correlations. Unfortunately, such a procedure was not feasible because of computational limitations, and we had to proceed by approximation, i.e. fitting a pdf via a PCC without considering the autoregressive processes. In particular, the computational limitations were due to the *tuning procedure* explained in appendix E. Therefore we used the best method possible to avoid underestimation of the risk uncertainties, but we underline that we used such an approximation.
The uncertainty of the return levels obtained via the observed data $h^{\text{obs}}$ are computed through propagating the parameter uncertainties of the GEV distribution fitted to the annual maxima of $h^{\text{obs}}$ (Figure 7). In particular, the fitted GEV distribution is a function of the parameters $\mu$ (location), $\sigma$ (scale) and $\eta$ (shape) (Coles, 2001). The GEV based return level $RL_t$ associated with the return period $t$ is a function of the three parameters $(\mu, \sigma, \eta)$ (Coles, 2001). We obtained the standard deviations of the three parameters $(\mu, \sigma, \eta)$, respectively $s_\mu$, $s_\sigma$, $s_\eta$, via the *gev.fit* function of the R package *ismev* (Heffernan and Stephenson,
2016). To estimate the standard deviation of the return level $RL_t$, we propagated the standard deviations of the three parameters $(\mu, \sigma, \eta)$ using the formula:

$$s_{\text{RL}_t} = \sqrt{\left(\frac{\partial RL_t}{\partial \mu}\right)^2 \cdot s_\mu^2 + \left(\frac{\partial RL_t}{\partial \sigma}\right)^2 \cdot s_\sigma^2 + \left(\frac{\partial RL_t}{\partial \eta}\right)^2 \cdot s_\eta^2} \tag{D1}$$

where $s_{RL}$ is the standard deviation of the return level $RL$. The final 95% interval of uncertainty of the return level $RT_t$ is obtained as $RT_t \pm 2s_{RL_t}$.

## Appendix E: Incorporation of the AR(1) in the unconditional model

Given a statistical model describing time series with serial correlations, to avoid an underestimation of the model uncertainties computed via bootstrap procedure, it is necessary to use a model which can reproduce the serial correlation. During the bootstrap procedure, simulating samples without serial correlation, and then re-fitting the model to each of them, would mean to assume that the data carry more information than they actually do. In fact, it is like the effective sample size of data with serial correlation is smaller than those without (Serinaldi, 2015). Here we introduce the procedure we used to build a multivariate statistical model that can represent the serial correlation and the marginal pdf of the variables, and the statistical dependencies between them. The steps taken follow below.

1. Fit a linear Gaussian autoregressive model of order 1, $AR(1)$:

$$Y_i(t) = c + \varphi Y_i(t-1) + \varepsilon_i(t) \tag{E1}$$

   on the $i^{th}$ marginal time series ($i = 1, 2, 3$), i.e. ($Y_{1_{Sea}}, Y_{2_{River}}, Y_{3_{River}}$). The chosen $AR(1)$ requires that the modelled variable is Gaussian distributed so, before the fit, we transformed the river variables via the $log_e$ function, which guarantees a more similar behaviour to the Gaussian. The observed sea variable was not transformed because it had already a behaviour similar to Gaussian.

2. Assured via the autcorrelation function (ACF) that $\varepsilon_i(t)$ has no longer a significant serial correlation, fit the joint pdf via PCCs on the residual variables ($\varepsilon_1, \varepsilon_2, \varepsilon_3$). We observe that the dependencies of these modelled pairs via PCCs, are not usual physical dependencies between the contributing variables (i.e. sea and river levels), but between their residuals with respect to the $AR(1)$ models.

3. Simulate the residuals ($\varepsilon_1^{sim}, \varepsilon_2^{sim}, \varepsilon_3^{sim}$) and plug into the $i^{th}$ autoregressive model. Finally, to get the simulated contributing variables $Y$, the river variables were transformed via the $exp$ function.

We observe here that when selecting the fitted pair-copulas and parameters for the residuals via the AIC, the simulated contributing variables $Y$ had a smaller dependence with respect to the observed. We therefore proceeded through a *tuning procedure*, i.e. we built a routine to automatically tune the parameters of the fitted families, requiring that the Kendall rank correlation coefficient among the contributing variables $Y$ were well simulated.

In Figure 12, the autocorrelation functions of the $Y^{obs}$ variables are compared with those of $Y^{sim}$ simulated from the fitted model. Because of the gaps in the $Y^{obs}$ time series, not all the observations are usable to compute the ACF (in particular the percentage of usable data decreases when increasing the Lag at which the ACF is computed). We therefore computed the ACF up to a Lag of about 25 days, which guarantees to use at least the 35% of data from the observed time series. Up to a Lag of

about 15 days, except for a very few cases with the variable $Y_{3_{River}}$ , the ACFs of the observed data are always inside the $95\%$ interval of the ACFs obtained from the fitted model.

We consider this result as satisfying because our target is to include the serial correlation of the contributing variables $Y$ into the model, and we can see that even for the variable $Y_{3_{River}}$ , the values of the ACFs have a significant serial correlation. Also, considering that the ACF is only slightly misrepresented for just one of the three time series, we argue that this is likely to have only a small effect on the final assessment of the model uncertainties.

[Figure 12 about here.]

## Appendix F: Brier score for extreme values

We employ the Brier score to assess the accuracy of the probabilistic predictions of the conditional model when predicting extreme values of the impact $h$ (e.g. Maraun, 2014). We defined an extreme of $h$ as a value larger than the 95-percentile of $h^{\text{obs}}$, the Brier score is:

$$BS = \frac{1}{N} \sum_{t=1}^{N} (p_{\text{t}} - o_{\text{t}})^2 \tag{F1}$$

where $p_{\text{t}}$ is the probability of getting an extreme value $h$ from the model at time $t$, while $o_{\text{t}}$ is 1 if $h^{\text{obs}}(t)$ is extreme and 0 otherwise. We get the value of $p_{\text{t}}$ through a Monte Carlo procedure.

The Brier skill score (BSS) measures the relative accuracy of the model under validation over a reference model, and is defined as:

$$BSS = 1 - \frac{BS}{BS_{ref}} \tag{F2}$$

where $BS_{ref}$ is the Brier score of the reference model. Here we consider the climatology of $h$ as the reference model, i.e. an empirical model such that $p_{\text{t}} = 0.05 \ \forall \ t$. A significant positive value of BSS indicates that when predicting extreme values, the model under validation is more accurate - according to the BS - than the reference model.

## Appendix G: Cross-validation procedure

To assess the quality of the conditional model, avoiding overfitting, we perform a 6-fold cross-validation. Therefore, the original sample of data $(\boldsymbol{X}, \boldsymbol{Y})$ is randomly partitioned into 6 equally sized subsamples. Of the 6 subsamples, 5 subsamples (the training data) are used in fitting the model that is then validated against the remaining subsample. For each training subsample we fit (1) new predictors $X$ for the contributing variables $Y$, (2) a new joint pdf $f_{\boldsymbol{Y}|\boldsymbol{X}}(\boldsymbol{Y}|\boldsymbol{X})$ and (3) a new h-function. For each validation subsample, we simulated $10^4$ realizations of the $\boldsymbol{Y}$ values through conditioning on the concurring predictors. Finally, by combining the simulations of each validation subsample, $10^4$ cross-validation time series of the contributing variables $Y$ and the impact $h$ are obtained.

*Author contributions.* DM had the initial idea for the study. EB and DM jointly developed the study with contributions by MW. EB developed the statistical model with contributions by IHH, DM and MV. EB carried out the analysis with contributions from DM and IHH. EB, DM and MW interpreted the results. EB wrote the paper with contributions from all other authors.

*Competing interests.* The authors declare that they have no conflict of interest.

5  *Acknowledgements.* E.B. received funding from the Volkswagen Foundation's CE:LLO project (Az.: 88468), which also supported project meetings. The authors would like to thank Prof. Arnoldo Frigessi for hosting them, and for fruitful discussions at the *Norwegian Computing Center*. E.B. would like to thank Colin Manning for the productive discussions, and contributions during the writing process. The authors would like to thank the anonymous reviewers for their valuable comments and suggestions which contributed to improving the quality of the paper. The data used for sea and river levels have been provided by the *Italian National Institute for Environmental Protection and Research*
10  *(ISPRA)* and *Arpae Emilia-Romagna*.

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

**List of Figures**

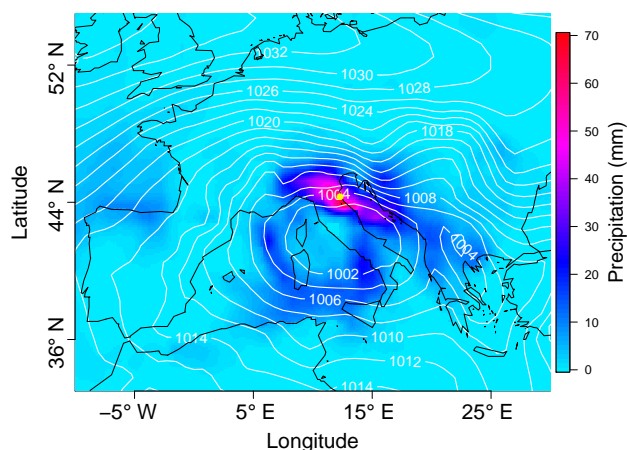

**Figure 1.** Sea level pressure and total precipitation on $6^{th}$ February 2015, when the coastal area of Ravenna (indicated by the yellow dot) was hit by a compound flooding.

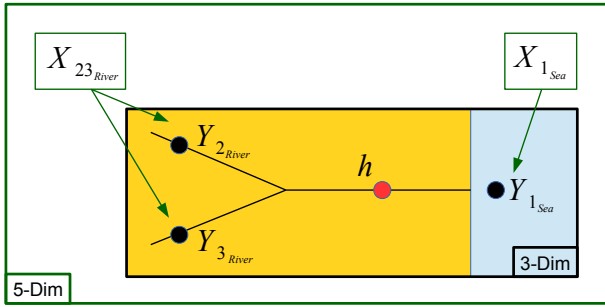

**Figure 2.** Hydraulic system for Ravenna catchment. The area affected by compound floods is marked by the red point. The impact is the water level $h$, which is influenced by the contributing variables $Y$, i.e. sea and river levels. The variables inside the black rectangle are used to develop the 3-dimensional (*unconditional*) model. The $X$ are the meteorological predictors driving the contributing variables $Y$, which are incorporated into the 5-dimensional (*conditional*) model.

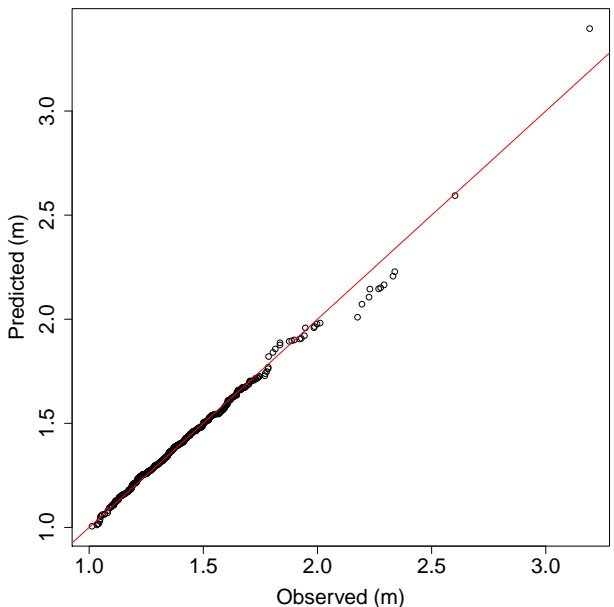

**Figure 3.** Q-Q plot between the observed impact (X-axis) and the modelled impact (Y-axis) from the regression model (equation (9)).

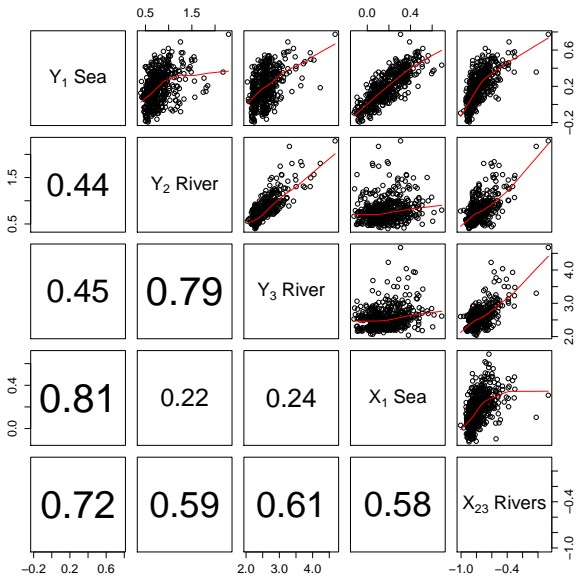

**Figure 4.** Scatter plots of predictands $Y^{obs}$ and predictors $X^{obs}$. The numbers are Spearman coefficient correlations. The red lines (computed via LOWESS, i.e. *Locally Weighted Scatterplot Smoothing*) is shown to better visualize the relationship between pairs (R Core Team, 2016).

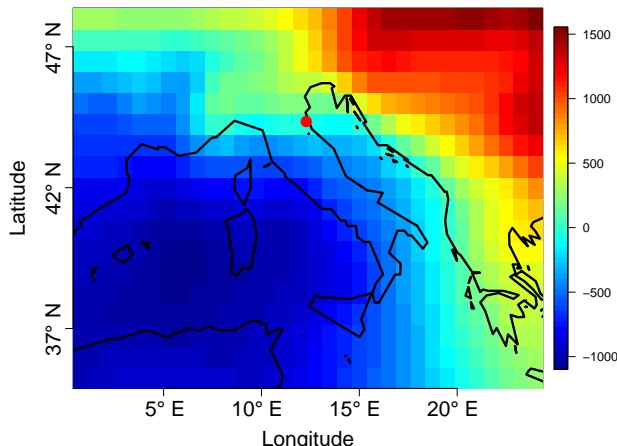

**Figure 5.** Regression Map $\boldsymbol{R}_{\text{MAP}}$ (equation (12)). The value of the regression map in the location $(i, j)$ is given by $R_{\text{MAP}}(i, j) = var(R_0)^{-1} \cdot cov(R_0, SLP_{i,j})$, where $R_0(t)$ is the residual of the barometric pressure effect obtained from the fit of the linear model $a_0 \, SLP_{Ravenna}(t) + d_0$ to $Y_{1_{\text{Sea}}}(t)$. The Regression map is equivalent to a 1-dimensional maximum covariance analysis (Widmann, 2005). The red dot indicates Ravenna.

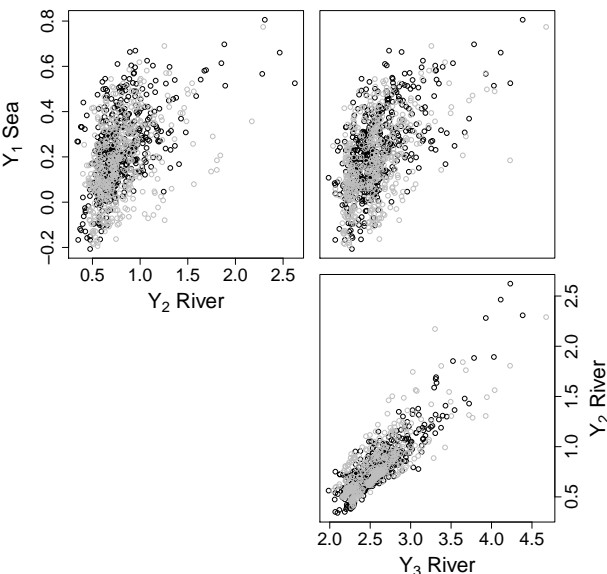

**Figure 6.** Scatter plots of observed (grey) against simulated (black) contributing variables $Y$. The simulated series are obtained via the 3-dimensional model (including the serial correlation), and have same length as the observed.

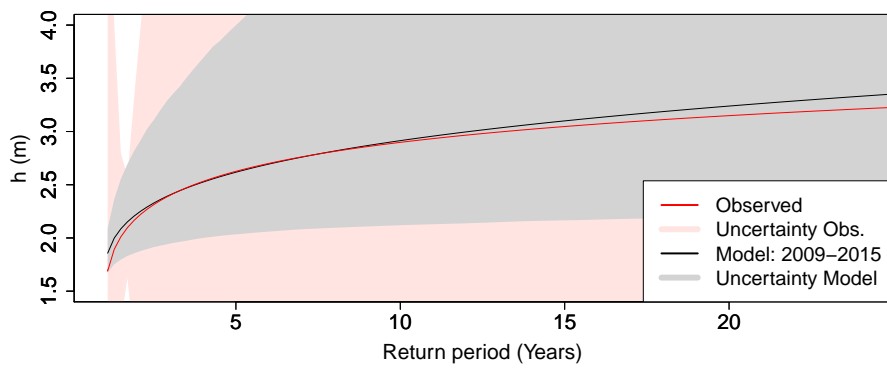

**Figure 7.** Unconditional model. Return levels of the impact $h$ with associated $95\%$ uncertainty intervals. The return level computed on $h^{\text{obs}}$ is shown in red (uncertainty shown in light red). The model based return level is shown in black (uncertainty is in grey).

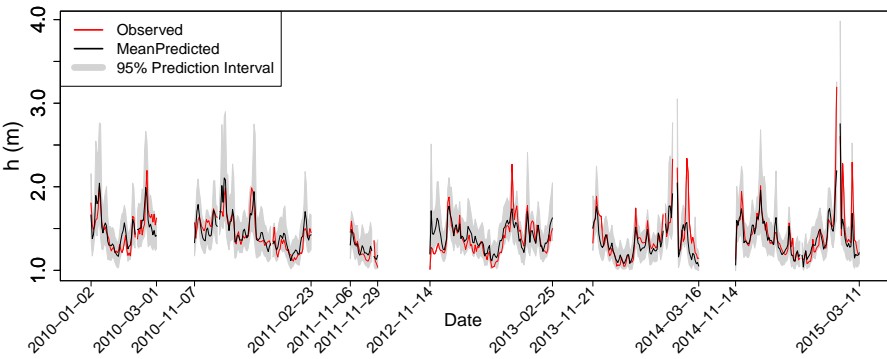

**Figure 8.** Validation time series of the conditional model obtained through 6-fold cross-validation. $h^{\mathrm{obs}}$ is shown in red. The average and $95\%$ prediction interval of $10^4$ simulated time series are respectively shown in black and grey.

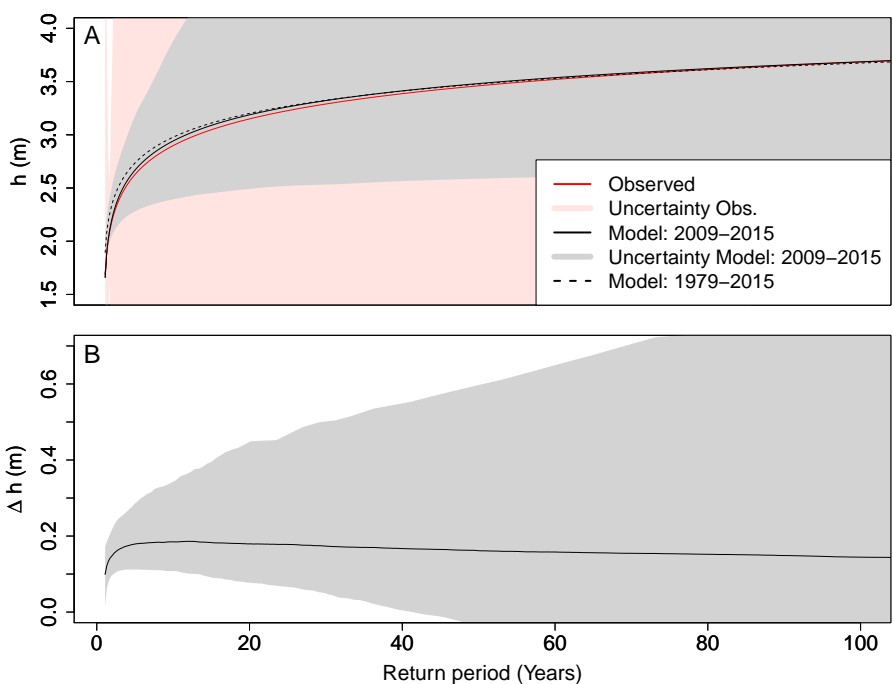

**Figure 9.** Conditional model. A: return levels of the impact $h$ with associated $95\%$ uncertainty intervals. The return level computed on $h^{\mathrm{obs}}$ is shown in red (uncertainty shown in light red). The model based return level computed for the period 2009-2015 (black) is based on $h^{\mathrm{sim}}$ values simulated for days where the observed data were available (uncertainty is shown in grey). The model based return level computed for the period 1979-2015 (black dashed) has uncertainty of similar amplitude to that of period 2009-2015 (not shown). B: difference between model based return level obtained when considering the realistic dependence between sea and river levels, and when assuming that they are independent. To make the dependencies between the sea and the river levels independent but keep the dependence between the two rivers, we shuffled the sea level data after each simulation, that guarantees random association between sea data and each of the rivers (e.g. Van den Hurk et al., 2015). The black line represents the median of the bootstrap samples.

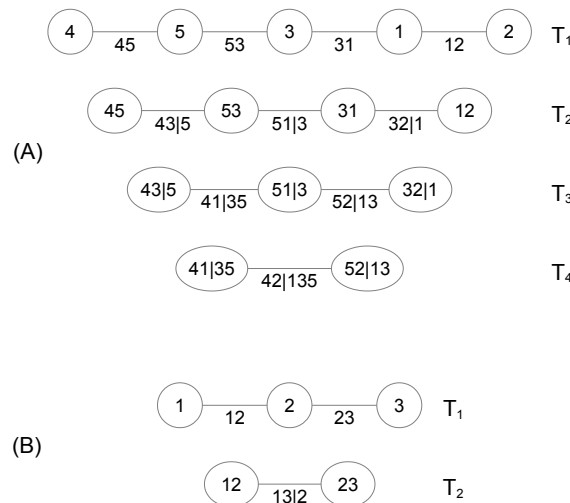

**Figure 10.** A: representation of the 5-dimensional D-vine in equation (6). There are 4 trees $(T_1, T_2, T_3, T_4)$, and 10 edges. Each edge represents a pair-copula density, and the label indicates the subscript of the corresponding copula. For example, the edge $43|5$ represents the copula density $c_{43|5}$. The decomposition of the joint pdf related to the represented vine is obtained by multiplying all the represented pair-copula densities (10 in this case) and the marginal pdfs of each variable. For more details see Aas et al. (2009). B: representation of the 3-dimensional vine in equation (B3). There are 2 trees ($T_1$ and $T_2$), and 3 edges.

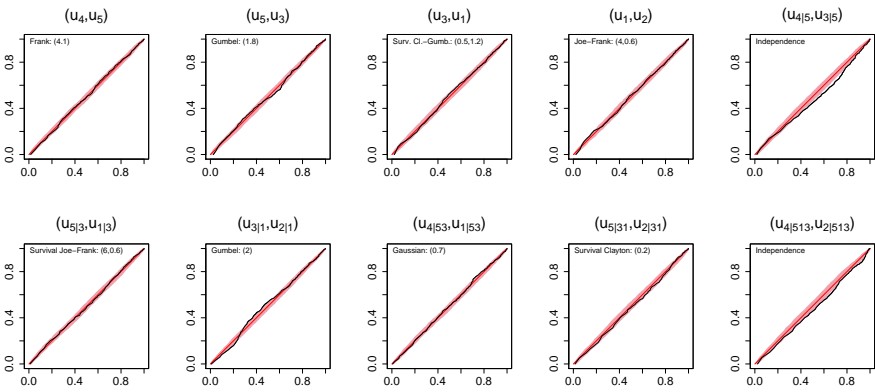

**Figure 11.** K-plots of the pair-copula families selected for the 5-dimensional model (name of the families and parameters are shown on the top-left of each plot). In abscissa the empirical K-function and in ordinate the K-function based on fitted copula. The $95\%$ confidence interval (shown in light red) is obtained from $10^4$ K-plots computed on simulated pairs (with same length as the observed data) from the selected pair-copula families.

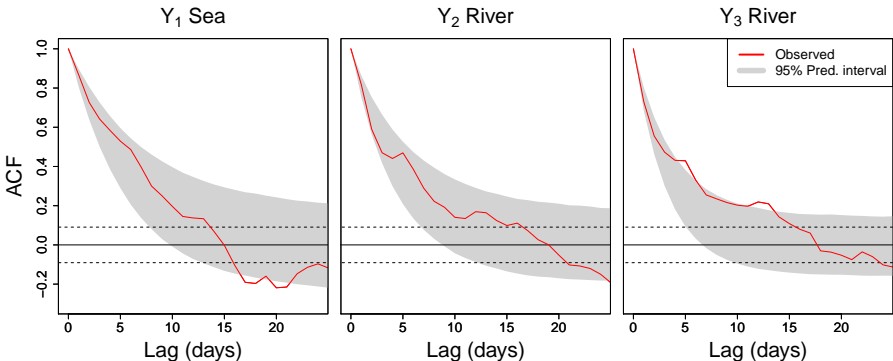

**Figure 12.** ACF of the observed time series (shown in red) against the ACF 95% confidence interval (grey) of the model (obtained through the Monte Carlo procedure). The dashed lines contain the 95% confidence interval defined by the ACF of a white noise process, i.e. outside this interval the ACF of the contributing variables $Y$ is significant.