# Peer review of "Multivariate Statistical Modelling of Compound Events via Pair-Copula Constructions: Analysis of Floods in Ravenna (Italy)"

_Hydrology and Earth System Sciences, 2016_

## Referee Comment (RC1) · Anonymous Referee #1 · 31 Jan 2017

**REVIEW REPORT**

**Journal:** Hydrol. Earth Syst. Sci.
**Paper:** HESS-2016-652
**Title:** Multivariate Statistical Modelling of Compound Events via Pair-Copula Constructions: Analysis of Floods in Ravenna
**Author(s):** Emanuele Bevacqua, Douglas Maraun, Ingrid Hobæk Haff, Martin Widmann, and Mathieu Vrac

**GENERAL COMMENTS.**

In my opinion, this paper represents a honest work dealing with a non-trivial problem. The Authors made a great effort to fix the model of interest, both at a physical and at a statistical level. I appreciated the fact that, beyond "advertising" the potential strengths of the model, the Authors also correctly mentioned (some of) its weaknesses, which might not be evident to unskilled practitioners.

In general, there does not exist a "perfect" model, especially if the problem is as difficult as the one investigated in this work. The approach suggested by the Authors is replete with model constraints and arbitrary assumptions, which are often (only) empirically justified. One may agree or disagree with the viewpoint and the *modus operandi* adopted by the Authors (and possibly suggest valuable alternatives), but overall I think that the procedures outlined in this paper look reasonable from a practical point of view, and the corresponding results could make sense (I also appreciated the discussion about uncertainties). Ultimately, I am in favor with having this work published, provided that the Authors fix the issues raised in the sequel.

**A note about Bibliography.** I was surprised that the following paper was not mentioned, since apparently it concerns the same Italian site (and about the same problem) investigated by the Authors: "Coastal flooding: A copula based approach for estimating the joint probability of water levels and waves" by Marinella Masina, Alberto Lamberti, and Renata Archetti; Coastal Engineering, Volume 97, March 2015, Pages 37-52. In addition, the reference to the 1997's book by Joe on copulas should be updated to the 2014's edition, and the reference to the 2007's book by Salvadori et al. should be corrected (missing co-authors). Finally, since the Authors use R packages, suitable references should be given in the Bibliography (not only in the text, it is useless!): it is the only "reward" that smart colleagues developing R free software do receive, and without a significant amount of citations, their Institutions will not give them anymore the possibility to go on producing such a bulk of procedures. Please, always give proper credits to whom deserve credits.

Some further comments follow below.

**SPECIFIC COMMENTS.**

**Page(s) 1, Title.**
> **Referee.** Usually, in international publications, if a site is mentioned in the title, then also the corresponding Country should be indicated: in turn, the Authors should write "Ravenna (Italy)".

**Page(s) 2, Line(s) 4–ff.**
> **Authors.** The impact of drought cannot be fully described by a single variable (e.g. Shiau et al. 2007)...

**Referee.** Here the Authors should also cite the seminal paper indicated below, where the usage of the Dynamic Return Period (i.e., the evolution of the joint RP along with the drought development) suggests mitigation strategies different from the univariate ones, traditionally used for assessing the risk (in agreement with some conclusions of the Authors).

C. De Michele, G. Salvadori, R. Vezzoli, and S. Pecora. "Multivariate assessment of droughts: frequency analysis and Dynamic Return Period". Water Resour. Res., 49(10):6985-6994, 2013.

**Page(s) 2, Line(s) 10.**

**Referee.** I am not sure that the adjective "systematic" is the proper one here (it could be deceiving). A systematic error "always goes in the same sense/direction", whereas the differences between univariate and multivariate results may not. Please use another adjective.

**Page(s) 6, Line(s) 8–ff.**

**Authors.** The impact $h$ of a CE can be formalized via an *impact-function*. . .

**Referee.** Essentially, this work adopts a multivariate Structural Approach, which has recently been well formalized in G. Salvadori, F. Durante, C. De Michele, M. Bernardi, and L. Petrella. "A multivariate Copula-based framework for dealing with Hazard Scenarios and Failure Probabilities". Water Resources Research, 53:3701-3721, 2016. Furthermore, useful guidelines for dealing with a multivariate Structural Approach in coastal/offshore engineering are given in G. Salvadori, F. Durante, G. R. Tomasicchio, and F. D'Alessandro. "Practical guidelines for the multivariate assessment of the structural risk in coastal and off-shore engineering". Coastal Engineering, 95:77-83, 2015. Actually, the structural approach discussed in these paper is practically the same as the one adopted by the Authors, but its mathematical/probabilistic foundation in terms of upper sets and suitable hazard scenarios is quite interesting and tickling, and may provide further (theoretical) support to the work of the Authors.

**Page(s) 6, Line(s) 8–ff.**

**Authors.** For instance, standard global and regional climate models do not simulate realistic runoff. . .

**Referee.** I am rather surprised by this sentence: could you please provide valuable references supporting such a strong claim?

**Page(s) 7, Line(s) 16.**

**Referee.** For the benefit of the unskilled readers and practitioners, the Authors should cite here some seminal books on copulas (e.g., Nelsen (2006), Salvadori et al. (2007), Joe (2014)), as well as some seminal papers like, e.g.,

C. Genest and A.C. Favre. "Everything you always wanted to know about copula modeling but were afraid to ask". J. Hydrol. Eng., 12(4):347-368, 2007.

G. Salvadori and C. De Michele. On the use of copulas in hydrology: theory and practice. J. Hydrol. Eng., 12(4):369-380, 2007.

**Page(s) 7, Line(s) 23.**

**Authors.** . . . it is possible to construct a valid joint pdf.

**Referee.** Prudentially, I would re-phrase the sentence as "in general, it is possible to construct a valid joint pdf, provided that suitable constraints are satisfied".

**Page(s) 7, Line(s) 25.**

**Referee.** Copulas do not "increase the number of available multivariate distributions", they only make it easier to play with more and more multivariate distributions: please re-phrase the sentence.

**Page(s) 8, Eq(s) 4–5.**

**Referee.** Before the equations, I would write "if the following limit exists and is non-zero".

**Page(s) 19, Line(s) 13–14.**

**Authors.** The accuracy of the estimated impact is very satisfactory...

**Referee.** Here, and throughout the paper, I would suggest to be more cautious about statements like the one reported above, especially given all the arbitrary assumptions/constraints introduced by the Authors, and the "visual" validations procedures. A sentence like "The accuracy of the estimated impact is empirically satisfactory..." may be more genuine.

**Page(s) 21, Line(s) 18–ff.**

**Referee.** Any way to show that the Simplifying Assumption (*simplified PCC*) does not affect (too much) the conclusions of this work?

**Page(s) 23–ff, Appendix C.**

**Referee.** There might be a lack of "objective" statistics here: diagnostic plots are often used instead of Goodness-of-Fit p-Values. Any way to get something better? I understand that computing p-Values using a Vine copula framework (even bootstrap ones) could be troublesome, but in general I do not like "visual" statistics (if not absolutely necessary or unavoidable).

Furthermore, the AIC is used to select the best Vine structure: as recently pointed out in the paper mentioned below, the AIC approach may not be a valuable solution when used for copulas. Instead, a cross-validation procedure (like, e.g., the one provided by the R package "copula" via the function "xvCopula") could be a better choice.

Steffen Grønneberg and Nils Lid Hjort. "The copula information criteria". Scandinavian Journal of Statistics, 41(2):436-459, 2014

---

## Referee Comment (RC2) · Anonymous Referee #2 · 20 Feb 2017

The paper by Bevacqua et al presents a very comprehensive assessment of a compound event by discussing floods in Ravenna. In a way the paper is a bit academic, as it cannot be expected to model return periods of annual maxima very accurately with only 6 years of data (e.g., section 5.3). This is reflected in the large uncertainties that prohibit clear conclusions such as that considering the dependence between drivers of the floods does not necessarily improve the predictions of more extreme return levels. However, the authors show that by including the relevant dependencies at much higher temporal scale, substantial reductions in uncertainty can be obtained. In particular the consequent propagation of uncertainties is a valuable contribution to the community working on compound events and other multivariate problems, and demonstrates how

large those uncertainties are in higher dimensional problems dealing with extremes. This is an often-ignored topic in this type of analysis. I thus consider the paper suitable for publication in HESS after some amendments which mostly refer to the presentation of the results as listed below.

Major comments: The structure of the paper could be improved quite a bit. Currently, large parts of the results are actually explanations of methods, selection and evaluation of model etc. For instance, the beginning of the result section would fit better into the methods section. Actual results are only presented from section 5.3 onward. And even later on, descriptions that belong to the methods part can be found throughout the text. Separating methods from results more clearly would improve the readability of the results section substantially. A discussion section missing although some points are discussed are in the conclusion section. I suggest renaming section 6 "Discussion and Conclusions" and also here more clearly separating the discussion from the conclusions.

Minor comments:

L6: "CEs" has not been defined yet as an acronym

L7: "downscaling of compound events"

L20: "obstructed" not sure what this word means here

L1: "recent report": the IPCC report was published 5 years ago, would not call that recent anymore

L 14: "Leonard et al., 2013": the year should be 2014

L5: avoid one-sentence paragraphs

L12 : this type of downscaling can be very useful, however, it can only be used at locations where at least some impact data is available and a model can be fitted since usually the fitted models are very context specific, which is also the case in this paper. I suggest omitting the sentences explaining the general usefulness of the downscaling of make it more specific for the applied case.

L3: I'm not convinced that the prior selection of parametric models generally reduces the uncertainty of the estimated quantity of interest. The uncertainty of selecting the right parametric model is just not considered in the final uncertainty estimates.

L25: I wouldn't say that copulas increase the number of available multivariate distributions. They only simplify the modelling of those.

L13: Maybe state that you will go through the 5 steps in detail in the next sections

L22: Maybe repeat the time period where impact data is available

L7: Is it reasonable to assume that the model has Gaussian noise?

L13: "Considering the two models. . .": "Omitting one of the variables as predictor leads to worse model performance, underlining the compound nature of the impact h"

L15: "The relative contribution. . .": omit and start the sentence with the part that comes afterward: "The sum of the relative contributions of the rivers. . ."

"red spot": "red dot"

Page 14:

L2: Specify which model you talk about

L13 and following: This should be moved to the methods section

L21: maybe also state the actual maximal value of h

L2: "is affected by uncertainties": "is affected by large uncertainties"

L3: this reads as if the model were not specifically designed for the floods in Ravenna. The discussed model can only be used for this specific case and location. For other places, new models would have to be designed and fitted to do downscaling (the number and location of rivers may be different, the mapping from the meteorological predictors X to Y might have a very different structure). Through this strong context dependence, compound events and models thereof are inherently difficult to generalize.

L14: delete "Environ. Res. Lett."

---

## Referee Comment (RC3) · Anonymous Referee #3 · 3 Mar 2017

The paper introduces a framework to assess compound flooding from storm surge and river discharge; the case study site is Ravenna in Italy where such an event caused major flooding in the recent past. The topic is a highly important one and falls into a very active research field. The authors propose a statistical modelling framework that exploits the copula theory by building pair copulas to model the 3 (in the stationary case) and 5 (including non-stationarity) dimensional problem at hand. The methods that are employed are state-of-the-art and in some places innovative. Bringing different types of statistical models together allows analyzing the complex problem of compound flooding under present-day, past, and future conditions paying particular attention to the uncertainties, which are often ignored in these kind of studies. I can see the conceptual

approach being adopted by other researchers and applied in different regions. I am in favor of publishing the manuscript with NHESS after some revision. I saw that the other reviewers already commented on two critical points, namely extending the cited literature and shifting text paragraphs around to better adhere to the structure that one would expect from the headers. Aside from that I list some comments below that should be addressed and are fairly minor. One thing that I was missing was the discussion of mean sea level rise, which is probably the most important driver for non-stationarity in the sea level component and as such in compound flood risk both over the past and in the future. I understand the model as it is would predict extreme events around the changing mean, this should be mentioned.

1-6 CE hasn't been defined

2-29 One typically cites those as Van den Hurk and Van den Brink (and puts them in the according place in the reference list)

5-28/29 Can you provide an example for that? It makes it easier for readers who are not experts on the different types of compound events.

8-25ff At this stage it was not clear to me how the selection was made for using this particular D-vine.

13-10ff Rivers flowing into the Adriatic are one contributor to the annual cycle that is not driven by barometric effects. Density changes due to temperature variations are probably also quite important.

15-5 Mention that this is not shown in the manuscript, at least I couldn't see it anywhere.

20-11 close bracket )

22-9 Merge Cooke (2001a, 2001b)

26-16 Repetition "depend on the dependence"

---

## Author Comment (AC1) · 24 Mar 2017

**Answers to comments from anonymous referees 1, 2 and 3**

March 24, 2017

We would like to thank the reviewers for carefully reading our manuscript and for their constructive comments which have considerably improved our manuscript. We agree with most suggestions and we implemented them in the revised version. Our major changes are related to the structure of the paper, as we agree with the referees that it could be improved.

Our intention is to provide a conceptual model to study generic compound events, and - based on this - study compound floods in Ravenna (Italy). With this in mind and referring to the referee comments, we modified the structure of the paper to help the reader to more easily read the manuscript. Following the referee suggestions, we moved the model development part from the "Results" section, and therefore we created a new section called "Model development". We show the structure of the paper at the end of the introduction as follows:

"The paper is organized as follows. The Ravenna case study is introduced in section 2. We present the conceptual model for compound events in section 3. Pair-copula constructions, i.e. the mathematical method we use to implement the model, is introduced in section 4. Based on the presented conceptual model for compound events, in section 5 we develop the model for compound floods in Ravenna. Results are presented in section 6 and conclusions are provided in section 7. More technical details can be found in the appendices. "

In doing so we emphasized the intention of introducing a conceptual model for compound events.

Moreover, following the referee comments, we extended the cited literature in the new version of the manuscript. Please find a detailed response below, where we quoted the referee comments in *Italic*.

**Referee 1**

**A note about Bibliography** *I was surprised that the following paper was not mentioned, since apparently it concerns the same Italian site (and about the same problem) investigated by the Authors: "Coastal flooding: A copula based approach for estimating the joint probability of water levels and waves" by Marinella Masina, Alberto Lamberti, and Renata Archetti; Coastal Engineering, Volume 97, March 2015, Pages 37-52. In addition, the reference to the 1997's book by Joe on copulas should be updated to the 2014's edition, and the reference to the 2007's book by Salvadori et al. should be corrected (missing co-authors). Finally, since the Authors use R packages, suitable references should be given in the Bibliography (not only in the text, it is useless!): it is the only "reward" that smart colleagues developing R free software do receive, and without a significant amount of citations, their Institutions will not give them anymore the possibility to go on producing such a bulk of procedures. Please, always give proper credits to whom deserve credits.*

We thank the referee for suggesting the paper by Masina et al. (2015). We included a reference to this paper in the section "Compound flooding in the coastal area of Ravenna" (pag. 4):

"As pointed out by Masina et al. (2015), natural and anthropogenic subsidences represent a threat for the coastal area of Ravenna, characterized by land elevation which are in many places below 2 m above mean sea level (Gambolati et al., 2002). The sea level inundation risk along the coast of Ravenna has been recently studied by Masina et al. (2015), who considered the joint effect of sea water level and significant wave height. "

About the reference corrections. We wrote:

- Joe, H.: Multivariate Models and Multivariate Dependence Concepts, Taylor Francis Ltd, United States, 2014.

- Salvadori, G., De Michele, C., Kottegoda, N.T., Rosso, R.: Extremes in nature: an approach using Copulas, Springer, Dordrecht, Netherlands., 2007.

Moreover, we inserted references to the R software and the packages as should be done:

- R Core Team (2016). R: A language and environment for statistical computing. R Foundation for Statistical Computing, Vienna, Austria. URL `https://www.R-project.org/`.

- Ulf Schepsmeier, Jakob Stoeber, Eike Christian Brechmann, Benedikt Graeler, Thomas Nagler and Tobias Erhardt (2016). VineCopula: Statistical Inference of Vine Copulas. R package version 2.0.5. `https://CRAN.R-project.org/package=VineCopula`.

- Heffernan, J. E. and Stephenson, A. G. (2016). ismev: An Introduction to Statistical Modeling of Extreme Values. R package version 1.41. `https://CRAN.R-project.org/package=ismev`.

Also, we added the reference to the R-package *CDVineCopulaConditional*, which has been just created, and contains the functions used in this paper to work with conditional vines:

- Emanuele Bevacqua (2017). CDVineCopulaConditional: Sampling from Conditional C- and D-Vine Copulas. R package version 0.1.0. `https://CRAN.R-project.org/package=CDVineCopulaConditional`.

**Specific comments**

1. *Page(s) 1, Title. Usually, in international publications, if a site is mentioned in the title, then also the corresponding Country should be indicated: in turn, the Authors should write "Ravenna (Italy)".*

   As suggested, we changed the title to "Multivariate Statistical Modelling of Compound Events via Pair-Copula Constructions: Analysis of Floods in Ravenna (Italy)".

2. *Page(s) 2, Line(s) 4–ff. Here the Authors should also cite the seminal paper indicated below, where the usage of the Dynamic Return Period (i.e., the evolution of the joint RP along with the drought development) suggests mitigation strategies different from the univariate ones, traditionally used for assessing the risk (in agreement with some conclusions of the Authors).*

   We added the suggested citation.

3. *Page(s) 2, Line(s) 10. I am not sure that the adjective "systematic" is the proper one here (it could be deceiving). A systematic error "always goes in the same sense/direction", whereas the differences between univariate and multivariate results may not. Please use another adjective.*

   We changed the sentence: "However this is not usually the case, and so would lead to systematic errors in the estimation of the risk associated with CEs." to "However this is not usually the case, and so would lead to misleading conclusions about the assessment of the risk associated with CEs."

4. *Page(s) 6, Line(s) 8–ff. Essentially, this work adopts a multivariate Structural Approach, which has recently been well formalized in Salvadori et al. (2016). Furthermore, useful guidelines for dealing with a multivariate Structural Approach in coastal/offshore engineering are given in Salvadori et al. (2015). Actually, the structural approach discussed in these paper is practically the same as the one adopted by the Authors, but its mathematical/probabilistic foundation in terms of upper sets and suitable hazard scenarios is quite interesting and tickling, and may provide further (theoretical) support to the work of the Authors.*

   We have inserted the following discussion at the end of the section.

   "In general, formalizing the impact $h$ of a CE as in step 1 - to then asses the risk of CE based on values of $h$ - corresponds to the *Structural Approach* (Salvadori et al. , 2015; Serinaldi, 2015; Volpi and Fiori, 2014), which has recently been formalized in Salvadori et al. (2016). Here, the

advantage of the general model we propose is that it allows for taking into account non-stationarity of the impact $h$ driven by temporal changes of the predictors $X$. Through the conditional pdf, the model allows for a realistic representation both of the dependencies between the $Y_i$, and of their marginal distributions. "

5. *Page(s) 6, Line(s) 8–ff. "For instance, standard global and regional climate models do not simulate realistic runoff". I am rather surprised by this sentence: could you please provide valuable references supporting such a strong claim?*

We added the following references:

Flato, G., J. Marotzke, B. Abiodun, P. Braconnot, S.C. Chou, W. Collins, P. Cox, F. Driouech, S. Emori, V. Eyring, C. Forest, P. Gleckler, E. Guilyardi, C. Jakob, V. Kattsov, C. Reason and M. Rummukainen, 2013: Evaluation of Climate Models. In: Climate Change 2013: The Physical Science Basis, 790–791, Contribution of Working Group I to the Fifth Assessment Report of the Intergovernmental Panel on Climate Change [Stocker, T.F., D. Qin, G.-K. Plattner, M. Tignor, S.K. Allen, J. Boschung, A. Nauels, Y. Xia, V. Bex and P.M. Midgley (eds.)]. Cambridge University Press, Cambridge, United Kingdom and New York, NY, USA, 2013.

Materia, S., Dirmeyer, P. A., Guo, Z., Alessandri, A. and Navarra, A.: The Sensitivity of Simulated River Discharge to Land Surface Representation and Meteorological Forcings, Journal of Hydrometeorology, 11(2), 334–351, doi:10.1175/2009jhm1162.1, 2010.

Tisseuil, C., Vrac, M., Lek, S. and Wade, A. J.: Statistical downscaling of river flows, Journal of Hydrology, 385(1-4), 279–291, doi:10.1016/j.jhydrol.2010.02.030, 2010.

6. *Page(s) 7, Line(s) 16. For the benefit of the unskilled readers and practitioners, the Authors should cite here some seminal books on copulas (e.g., Nelsen (2006), Salvadori et al. (2007), Joe (2014)), as well as some seminal papers like, e.g., Genest and Favre (2007) and Salvadori and De Michele (2007).*

We added the suggested references.

7. *Page(s) 7, Line(s) 23. Authors: it is possible to construct a valid joint pdf. Referee: Prudentially, I would re-phrase the sentence as "in general, it is possible to construct a valid joint pdf, provided that suitable constraints are satisfied"..*

We changed the sentence: "In fact, inserting any existing family for the marginal pdfs and copula density into eq. (3), it is possible to construct a valid joint pdf." to "In fact, inserting any existing family for the marginal pdfs and copula density into eq. (3), it is possible to construct a valid joint pdf, provided that suitable constraints are satisfied"

8. *Page(s) 7, Line(s) 25. Copulas do not "increase the number of available multivariate distributions", they only make it easier to play with more and more multivariate distributions: please re-phrase the sentence.*

We changed the sentence to "Copulas therefore make it easy to construct a wide range of multivariate parametric distributions."

9. *Page(s) 8, Eq(s) 4–5. Before the equations, I would write "if the following limit exists and is non-zero".*

We did rephrase as:

Mathematically, given two random variables $Y_1$ and $Y_2$ with marginal CDFs $F_1$ and $F_2$ respectively, they are *upper tail dependent* if the following limit exists and is non-zero:

$$\lambda_U(Y_1, Y_2) = \lim_{u \to 1} P(Y_2 > F_2^{-1}(u) | Y_1 > F_1^{-1}(u)) \qquad (1)$$

where $P(A|B)$ indicates the generic conditional probability of occurrence of the event $A$ given the event $B$. Similarly, the two variables are *lower tail dependent* if:

$$\lambda_L(Y_1, Y_2) = \lim_{u \to 0} P(Y_2 < F_2^{-1}(u) | Y_1 < F_1^{-1}(u)) \qquad (2)$$

exists and is non-zero.

10. *Page(s) 19, Line(s) 13–14. Authors: The accuracy of the estimated impact is very satisfactory. Referee: Here, and throughout the paper, I would suggest to be more cautious about statements like the one reported above, especially given all the arbitrary assumptions/constraints introduced by the Authors, and the "visual" validations procedures. A sentence like "The accuracy of the estimated impact is empirically satisfactory. . . " may be more genuine.*

We agree with the referee. However we argue that it may be better to replace the sentence "The accuracy of the estimated impact is very satisfactory" with "The accuracy of the estimated impact appears satisfactory" instead of using "The accuracy of the estimated impact is empirically satisfactory". This choice is adopted as we argue that all types of comparisons between model output and observations are "empirical", i.e. empirically-based. Therefore our choice is even outlining more the point of the referee.

Moreover we changed the sentence "The model successfully captures the overall temporal evolution of the impact (Figure 8),..." to "The overall temporal evolution of the impact appears successfully captured by the model (Figure 8),..."

11. *Page(s) 21, Line(s) 18–ff. Any way to show that the Simplifying Assumption (simplified PCC) does not affect (too much) the conclusions of this work?*

As we pointed out in the paper "The simplified PCC may be a rather good approximation, even when the simplifying assumption is far from being fulfilled by the actual model (Hobæk Haff et al., 2010; Stöber et al., 2013)." However, to verify the goodness of the assumption, we did fit the models in equation (B3) and (B4) without assuming simplified PCC.

The top copula in equation (B3) is $c_{13|2}(u_{1|2}, u_{3|2})$, which is $c_{13|2}(u_{1|2}, u_{3|2}; u_2)$ when not assuming the simplified PCC. This is a survival Gumbel copula which has one parameter $\theta$. The parameter of the copula appears to be an increasing function of the conditioning variable $u_2$, that ranges from $\theta = 1.06$ to $\theta = 1.23$, with a mean equal to the value estimated for

the simplified vine ($\theta = 1.13$). However, this parameter range does not change the copula that much. For example, the Kendall Tau coefficient range from 0.06 ($\theta = 1.06$) to 0.18 ($\theta = 1.23$), while that corresponding to the value estimated for the simplified vine is 0.12 ($\theta = 1.13$). This means that the bias of the Kendall Tau coefficient of the simplified copula is in the range $[-0.06; +0.06]$.

We argue that potential differences due to the simplified assumption are averaged out during the simulations from the vine. This vine is modelling the residuals $\varepsilon_i$ of the AR(1) models. Extreme values of the impact $h$ are not related with particular values of $\varepsilon_2$. Therefore when simulating extreme values of $h$, all the values of $\varepsilon_2$ may coincide with extremes of $h$. So, when simulating extreme values of h, the dependence of the copula $c_{13|2}$ ($i := \varepsilon_i$) assumed with the simplified assumption is slightly smaller or bigger than the value that would be adopted without simplifying assumption. But this effect should be statistically averaged out. Moreover, the bias in the "Kendall tau" is small (in the range $[-0.06; +0.06]$) and therefore should not affect too much the final result in any case.

For the model in equation (B4), the top copula is $c_{32|1}(u_{3|1}, u_{2|1})$ (i.e. $c_{32|1}(u_{3|1}, u_{2|1}; u_1)$ when not assuming simplified PCC). This copula is a type 1 Twan, which has two parameters. For simplicity, we did estimate the parameters of the copula under non-simplified assumption for one parameter. That is, we estimated the first parameter as a function of the conditioning variable, and kept the second one fixed. As result, the parameter is an erratic function jumping up and down between 2.06 and 2.75, with a mean equal to the value estimated for the simplified vine. We tried to increase the bandwidth for $u_1$ when fitting the parameter, but we did still find this erratic behaviour. Considering such behaviour around the same value, which we interpret as a noise, we think that it may be even better to keep the parameter fixed at the mean value. This is actually what we do when making the simplifying assumption.

12. *Page(s) 23–ff, Appendix C. There might be a lack of "objective" statistics here: diagnostic plots are often used instead of Goodness-of-Fit p-Values. Any way to get something better? I understand that computing pValues using a Vine copula framework (even bootstrap ones) could be troublesome, but in general I do not like "visual" statistics (if not absolutely necessary or unavoidable).*

We argue that diagnostic plots may often offer additional useful information to formal good goodness of fit tests. For example, when using K-plots, it is possible to get separate information for different quantiles (when using Goodness-of-Fit p-Values, the information is projected in only one number). Here we computed confidence intervals for the K-plots, which give additional information to evaluate the goodness of the fit.

We performed goodness-of-fit tests (we used the Cramér-von Mises test based on the Kendall function) for the 10 copulas in the 5-dimensional vine. The p-values from the tests are:

Level 1: Copula 1: 0.76, Copula 2: 0.08, Copula 3: 0.77, Copula 4: 0.36

Level 2: Copula 1 (independence copula): 0.00028, Copula 2: 0.24, Copula 3: 0.0

Level 3: Copula 1: 0.19, Copula 2: 0.012

Level 4: Copula 1 (independence copula): 0.00088

Therefore the null hypothesis that the data come from the chosen copulas is not rejected at 99% confidence level, except from the two independence copulas on level 2 and 4, and Copula 3 on level 2. However, as explained in "Appendix C", the two independence copulas were set because the selection algorithm had chosen copulas with a slightly negative dependence, which did not make sense physically. It is therefore not surprising that the hypothesis of independence is rejected. For Copula 3 on level 2, it would be difficult to find a parametric copula that gives a better fit than the used. To be sure that a better choice cannot be done for such copula, we tried fitting all of the parametric copulas in use. However, when we performed the Cramér-von Mises test to the 8 best families (according to AIC), we always got p-values=0. Therefore we kept the parametric copula which was selected according to the AIC.

*Furthermore, the AIC is used to select the best Vine structure: as recently pointed out in the paper mentioned below, the AIC approach may not be a valuable solution when used for copulas. Instead, a cross-validation procedure (like, e.g., the one provided by the R package "copula" via the function "xvCopula") could be a better choice.*

*Steffen Grønneberg and Nils Lid Hjort. "The copula information criteria". Scandinavian Journal of Statistics, 41(2):436-459, 2014*

Grønneberg and Hjort (2014) suggest - in the bivariate context where they work - to replace the cross validation procedure with the xv-CIC, for large n. In our case n $\sim$ 500, and we are in a multivariate case ($d > 2$), therefore we argue that it is justified to replace the cross validation procedure with the xv-CIC. However, Jordanger and Tjøstheim (2014) show that only minor differences are observed when using xv-CIC instead of AIC (in particular they show such statement for n=500). Based on this argumentation, we think that it is reasonable to use the AIC.

Lars Arne Jordanger and Dag Tjøstheim, Model selection of copulas: AIC versus a cross validation copula information criterion, Statistics and Probability Letters, Volume 92, 249–255, 2014.

**Referee 2**

***Major comments:*** *The structure of the paper could be improved quite a bit. Currently, large parts of the results are actually explanations of methods, selection and evaluation of model etc. For instance, the beginning of the result section would fit better into the methods section. Actual results are only presented from section 5.3 onward. And even later on, descriptions that belong to the methods part can be found throughout the text. Separating methods from results more clearly would improve the readability of the results section substantially.*

We agree with the reviewer comment. Therefore we moved the first part of the result section to the new section "Model development". And the new section "Results" starts with the old section 5.3. Also, we tried moving other methodology descriptions to the method sections.

*A discussion section missing although some points are discussed are in the conclusion section. I suggest renaming section 6 "Discussion and Conclusions" and also here more clearly separating the discussion from the conclusions.*

We changed the title of the section to "Discussion and Conclusions". Additionally, we tried to emphasize the separation between the first part of this section (the general discussion about the conceptual model) and the second one (the discussion of conceptual model application to compound floods in Ravenna).

***Minor comments:***

1. *Page 1, L6: "CEs" has not been defined yet as an acronym*

   We did replace "CEs" with "compound events"

2. *Page 1, L7: "downscaling of compound events"*

   To avoid the repetition, we replaced the sentences: "Moreover, this model provides multivariate statistical downscaling of compound events. Downscaling of compound events is required to extend their risk assessment to the past or future climate, where climate models either do not simulate realistic values of the local variables driving the events, or do not simulate them at all." with "Moreover, this model enables multivariate statistical downscaling of compound events. Downscaling is required to extend the compound events risk assessment to the past or future climate, where climate models either do not simulate realistic values of the local variables driving the events, or do not simulate them at all."

3. *Page 1, L20: "obstructed" not sure what this word means here*

   We rephrased as: "Alongside the storm surge, large amounts of precipitation fell in the surrounding area causing high values of discharge in small rivers near the coast. These river discharges were partially obstructed from draining into the sea by the storm surge, which then contributed to major flooding along the coast. "

4. *Page 2, L1: "recent report": the IPCC report was published 5 years ago, would not call that recent anymore*

   We removed "recent".

5. *Page 2,L 14: "Leonard et al., 2013": the year should be 2014*

   We corrected the year.

6. *Page 6, L5: avoid one-sentence paragraphs*

   We unified the one-sentence paragraph with the next paragraph. Now, this looks like the following:

   "Our non-stationary multivariate statistical model consists of three components: the contributing variables $Y_i$, including a model of their dependence structure, the impact $h$, and meteorological predictors $X_j$ of the contributing variables. The contributing variables $Y_i$ and their multivariate dependence structure define the CE. For instance, in case of compound floods, these are runoff and sea level. The impact $h$ of a CE can be formalized via an *impact-function* $h = h(Y_1, ..., Y_n)$. In the case of compound flooding, we define the river gauge level in Ravenna as impact, but in principle it can be any measurable variable such as, e.g, agricultural yield or economic loss. The predictors $X_j$ provide insight into the physical processes underlying CEs, including the temporal variability of CEs, and can be used to statistically downscale CEs (Maraun et al., 2010).

7. *Page 6, L12 : this type of downscaling can be very useful, however, it can only be used at locations where at least some impact data is available and a model can be fitted since usually the fitted models are very context specific, which is also the case in this paper. I suggest omitting the sentences explaining the general usefulness of the downscaling of make it more specific for the applied case.*

   We agree with the referee that the model can be applied only when appropriate observations are available for the calibration, and therefore we explain this better. However, as this section introduces the conceptual model for modelling generic compound events, we prefer to keep the discussion at a general level.

   We modified the sentence as:

   "The predictors $X_j$ provide insight into the physical processes underlying CEs, including the temporal variability of CEs, and can be used to statistically downscale CEs when the variables $Y$ and the impact $h$ are available (e.g. Maraun et al., 2010). "

   Also, at the end of the section we added the following:

   When the variables $Y$ are available but not the impact $h$, the model can still be used to only estimate the variables $Y$. This may be useful when assessing the risk of CEs through, e.g., multivariate return periods of the contributing variables $Y$ (e.g. Graeler et al., 2016, 2013; Salvadori et al., 2016, 2011; Wahl et al., 2015; Aghakouchak et al., 2014; Saghafian and Mehdikhani, 2013; Shiau et al., 2007; Shiau, 2003). Moreover, it may happen that the impact $h$ is available, but the variables $Y$ are not. In this case the model may still be used in the form $f_{h|\vec{X}}(h|\vec{X})$ to directly estimate the impact $h$, based on the conditional joint pdf of the impact $h$, given the predictors $X$. In this case, depending on the physical system, it may be more or less complicated to calibrate the predictors. Also, we observe that equation (1) is general and a possibility for estimating the impact

would be to use the conditional joint pdf $f_{h|\vec{Y}}(h|\vec{Y})$. Such an approach may be useful for cases where complex relations exist between the impact $h$ and the variables $Y$, and therefore it may be difficult to implement, e.g., a proper regression model to describe the impact $h$.

8. *Page 7, L3: I'm not convinced that the prior selection of parametric models generally reduces the uncertainty of the estimated quantity of interest. The uncertainty of selecting the right parametric model is just not considered in the final uncertainty estimates.*

    We actually agree with this comment. After the original sentences:

    "An advantage of using a parametric statistical model is that this constrains the dependencies between the contributing variables, as well as their marginal distributions, and thereby reduces their uncertainties with respect to empirical estimates (Hobæk Haff et al., 2015). Such a reduction as above in turn reduces the uncertainty in the estimated physical quantity of interest, like the impact of the CE.",

    we added the following:

    "However, the uncertainty reduction depends on the choice of a proper parametric model, in particular when modelling the tail of a univariate or multivariate distribution. "

    We added a similar sentence in the introduction, after a similar statement to that criticized by the referee.

9. *Page 7, L25: I wouldn't say that copulas increase the number of available multivariate distributions. They only simplify the modelling of those.*

    We changed the sentence "Copulas therefore increase the number of available multivariate distributions." to "Copulas therefore make it easy to construct a wide range of multivariate parametric distributions."

10. *Page 9, L13: Maybe state that you will go through the 5 steps in detail in the next sections*

    We modified the introduction to the steps as suggested: "Below we show the steps we follow to study compound floods in Ravenna, based on the conceptual model described in section 3. We will go through these steps in detail in the next sections. "

11. *Page 9, L22: Maybe repeat the time period where impact data is available*

    We replaced the sentence: "Here, we use the model to extend the multivariate time series $\vec{Y}(t)$ to the past (period 1979-2015), when only $\vec{X}(t)$ is available." with "Here, we use the model (calibrated to the period 2009-2015) to extend the multivariate time series $\vec{Y}(t)$ to the past (period 1979-2015), when only $\vec{X}(t)$ is available."

12. *Page 10, L7: Is it reasonable to assume that the model has Gaussian noise?*

    The choice of the model is reasonable as, even if slightly skewed, the distribution looks normal. The qq-plot of the fitted distribution appears satisfying (also when compared with those obtained for other distributions, i.e. t, logistic and cauchy).

13. *Page 10, L13: "Considering the two models. . .": "Omitting one of the variables as predictor leads to worse model performance, underlining the compound nature of the impact h"*

    We followed the referee's suggestion.

14. *Page 10, L15: "The relative contribution. . .": omit and start the sentence with the part that comes afterward: "The sum of the relative contributions of the rivers. . ."*

    We followed the suggestion from the referee.

15. *Page 13 "red spot": "red dot"*

    We replace "spot" with "dot" twice in the paper.

16. *Page 14, L2: Specify which model you talk about*

    We replaced the sentence "This model reproduces the joint pdf of the contributing variables..." with "The stationary model reproduces the joint pdf of the contributing variables..."

17. *Page 13, L13: and following: This should be moved to the methods section*

    This part was moved to the method section, namely to the "Model development" section.

18. *Page 16, L21: maybe also state the actual maximal value of h*

    We inserted the maximal value of h as: "The expected return period of the highest compound flood observed $(3.19m)$, computed over the period 2009-2015, is 20 years".

    Similarly, we inserted such value also in the result section.

19. *Page 17, L2: "is affected by uncertainties": "is affected by large uncertainties"*

    We followed the suggestion. We changed the sentence "However, this value is affected by uncertainties..." to "However, this value is affected by large uncertainties..."

20. *Page 18, L3: this reads as if the model were not specifically designed for the floods in Ravenna. The discussed model can only be used for this specific case and location. For other places, new models would have to be designed and fitted to do downscaling (the number and location of rivers may be different, the mapping from the meteorological predictors X to Y might have a very different structure). Through this strong context dependence, compound events and models thereof are inherently difficult to generalize.*

    At this point we refer to the conceptual model for compound events. We made this clear in the conclusion, writing:

    "The conceptual model is particularly useful to downscale large scale predictors from climate models..."

21. *Page 32, L14: delete "Environ. Res. Lett."*

    We did correct the reference.

**Referee 3**

*The paper introduces a framework to assess compound flooding from storm surge and river discharge; the case study site is Ravenna in Italy where such an event caused major flooding in the recent past. The topic is a highly important one and falls into a very active research field. The authors propose a statistical modelling framework that exploits the copula theory by building pair copulas to model the 3 (in the stationary case) and 5 (including non-stationarity) dimensional problem at hand. The methods that are employed are state-of-the-art and in some places innovative. Bringing different types of statistical models together allows analyzing the complex problem of compound flooding under present-day, past, and future conditions paying particular attention to the uncertainties, which are often ignored in these kind of studies. I can see the conceptual approach being adopted by other researchers and applied in different regions. I am in favor of publishing the manuscript with NHESS after some revision. I saw that the other reviewers already commented on two critical points, namely extending the cited literature and shifting text paragraphs around to better adhere to the structure that one would expect from the headers. Aside from that I list some comments below that should be addressed and are fairly minor. One thing that I was missing was the discussion of mean sea level rise, which is probably the most important driver for non-stationarity in the sea level component and as such in compound flood risk both over the past and in the future. I understand the model as it is would predict extreme events around the changing mean, this should be mentioned.*

We agree that in general the mean sea level rise (SLR) has to be considered when assessing the evolution of compound floods risk, in particular for risk assessment in the future, as an important SLR is expected for the Mediterranean Sea during the next century. The SLR would be easily included in our model through adding an extra term in the definition of the Sea level predictor $X_{1_{Sea}}$, or directly adding the SLR term to the simulated Sea level $Y_{1_{Sea}}^{sim}$.

However, SLR was not considered in this analysis because during the analysed period (1979-2015) it was negligible in the North Adriatic Sea ($\sim 0.8mm/year$). The observed mean sea level rise rate has been: $(0.58 \pm 0.20)mm/year$ at Rovinj, Croatia (based on data from the period 1955-2009); $(0.84 \pm 0.53)mm/year$ at Trieste, Italy (based on 1970-2011); $(0.97 \pm 0.36)mm/year$ at Bakar, Croatia (based on 1930-2009) (https://tidesandcurrents.noaa.gov/sltrends/sltrends.html). We observe that Trieste is located on a stable area, i.e. where subsidence and uplift are negligible, therefore the observed SLR in Trieste is attributable to the eustatic rise only (Carbognin et al., 2011). An increase of $\sim 0.8mm/year$ (the mean of listed above) would correspond to an increase of $\Delta \sim 2.9cm$ during the 36 analysed years (period 1979-2015). This value is small when compared with the observed range of variation of the sea level ($\sim 100cm$). Moreover the total variation of the impact $h$ due to such SLR ($\Delta \sim 2.9cm$) would be $\Delta \cdot a_1 \sim 2.6cm$ ($a_1$ is defined in equation (9) of the discussion paper), which is small compared with the total range of variation of $h$ ($\sim 220cm$).

We add that, in general, subsidence represent a threat for the coastal area of Ravenna (Masina et al. (2015), Carbognin et al. (2011)), therefore this may need to be considered. However, although Ravenna has experienced a relative

sea level rise (RSLR) rate of $8.5mm/year$ in the last century, this has been negligible during the analysed period (Carbognin et al., 2011).

In response to the reviewer's comment, however, we have added the following sentence after defining the sea level predictor:

"Moreover, we will not consider long-term sea level rise because its influence on both sea and impact $h$ level variations is negligible over the considered period (the observed rate of sea level rise in the North Adriatic Sea has been $\sim 0.8mm/year$ (NOAA, Tides & Currents)). Also the relative sea level rise has been negligible over the considered period (Carbognin et al., 2011)"

Carbognin, L., Teatini, P., Tosi, L., Strozzi, T. and Tomasin, A.: Present Relative Sea Level Rise in the Northern Adriatic Coastal Area, In: Coastal and marine spatial planning. Marine Research at CNR, DTA/06 . CNR - Dipartimento Scienze del Sistema Terra e Tecnologie , Roma, pp. 1147-1162, 2011.

Masina, M., Lamberti, A. and Archetti, R.: Coastal flooding: A copula based approach for estimating the joint probability of water levels and waves, Coastal Engineering, 97, 37–52, doi:10.1016/j.coastaleng.2014.12.010, 2015.

**Specific comments**

1. *1-6 CE hasn't been defined*

   We did replace "CEs" with "compound events"

2. *2-29 One typically cites those as Van den Hurk and Van den Brink (and puts them in the according place in the reference list)*

   We did used "Van den Hurk" and "Van den Brink" (also for other references in the reference list). Moreover, we put them at the letter V in the reference list.

3. *5-28/29 Can you provide an example for that? It makes it easier for readers who are not experts on the different types of compound events.*

   We think that an example will help the reader. We replaced the sentence: "For example, there can be a mutual reinforcement of one variable by the other and vice versa due to system feedbacks (Seneviratne et al., 2012)" with "There can be a mutual reinforcement of one variable by the other and vice versa due to system feedbacks, e.g. the mutual enhancement of droughts and heat waves in transitional regions between dry and wet climates (Seneviratne et al., 2012). "

4. *8-25ff At this stage it was not clear to me how the selection was made for using this particular D-vine.*

   We added the sentence "Details about the selection procedure of the vine (eq. (6)) are given in appendices B2 and C ..." just after introduced the vine.

5. *13-10ff Rivers flowing into the Adriatic are one contributor to the annual cycle that is not driven by barometric effects. Density changes due to temperature variations are probably also quite important.*

We wrote the sentence: "This harmonic term could be driven by the annual hydrological cycle (Tsimplis et al., 1994), i.e. due to cyclic runoff of rivers which flow into the Adriatic sea, or due to density variations of the sea water (due to the annual cycle of water temperatures). "

6. *15-5 Mention that this is not shown in the manuscript, at least I couldn't see it anywhere.*

   We inserted it as: "For example, the amplitude of the 95% confidence interval of the 20-years return level is underestimated by about 50% (not shown)."

7. *20-11 close bracket )*

   Thanks, we inserted the missing bracket ")".

8. *22-9 Merge Cooke (2001a, 2001b)*

   We did merge the citations.

9. *26-16 Repetition "depend on the dependence"*

   We replaced the sentence "In particular, we observed that the uncertainties depend on the dependence values between the modelled pairs (not shown)." with "In particular, we observed that the uncertainties are also controlled by the dependence values between the modelled pairs (not shown)."